# Ordovician opabiniid-like animals and the role of the proboscis in euarthropod head evolution

Stephen Pates [1], Joseph P. Botting [2,3], Lucy A. Muir [3] & Joanna M. Wolfe[4]

A crucial step in the evolution of Euarthropoda (chelicerates, myriapods, pancrustaceans) was the transition between fossil groups that possessed frontal appendages innervated by the first segment of the brain (proto-cerebrum), and living groups with a protocerebral labrum and paired appendages innervated by the second brain segment (deutocerebrum). Appendage homologies between the groups are controversial. Here we describe two specimens of opabiniid-like euarthropods, each bearing an anterior proboscis (a fused protocerebral appendage), from the Middle Ordovician Castle Bank Biota, Wales, UK. Phylogenetic analyses support a paraphyletic grade of stem-group euarthropods with fused protocerebral appendages and a posterior-facing mouth, as in the iconic Cambrian panarthropod *Opabinia*. These results suggest that the labrum may have reduced from an already-fused proboscis, rather than a pair of arthropodized appendages. If some shared features between the Castle Bank specimens and radiodonts are considered convergent rather than homologous, phylogenetic analyses retrieve them as opabiniids, substantially extending the geographic and temporal range of Opabiniidae.

Exceptional Palaeozoic deposits preserving fossils of soft-bodied organisms, such as the renowned Cambrian Chengjiang and Burgess Shale biotas, continue to offer critical insights into the diversity, ecology, and evolution of early animals[1–9]. The arrangement of many of these iconic and unusual early animals into stem lineages leading to extant crown groups has been particularly important for resolving the origin of fundamental features defining crown groups[10–12]. Within Euarthropoda (chelicerates, myriapods and pancrustaceans), a phylum estimated to comprise more than three-quarters of all living animal species[13], stem group fossils record evolution from a lobopodian-like ancestor with paired appendages innervated by the protocerebrum, the neuromere of the anteriormost segment of the brain[11,12,14]. Variation in protocerebral appendage morphology allowed lower stem-group euarthropods to explore numerous ecological niches during the Palaeozoic. Radiodonts (relatives of *Anomalocaris*) represent one of

the most notable examples, occupying ecological niches from filter feeders to apex predators[15–19].

Hypothesised relationships of extant and fossil euarthropods often rely heavily on interpretations of head segmentation and homologies between anterior appendages; however this topic remains controversial[14,20–23]. Developmental, morphological and neurological data support the interpretation that protocerebral appendages transformed through the euarthropod stem lineage, from paired annulated appendages in gilled lobopodians such as *Kerygmachela*, to a fused proboscis in opabiniids, to the arthropodized and sclerotized appendages of radiodonts[3,14,20,24,25], and subsequently the fused labrum as seen in nearly all extant euarthropods[7,14,26] (but see e.g. refs. 27–29 for alternative views on the homology radiodont frontal appendages). However, recent discoveries and reinterpretations of palaeoneurological data have called into question the evidence for protocerebral

[1]Department of Zoology, University of Cambridge, Cambridge, UK. [2]Nanjing Institute of Geology and Palaeontology, Chinese Academy of Sciences, Nanjing, China. [3]Department of Natural Sciences, Amgueddfa Cymru—National Museum Wales, Cardiff, UK. [4]Museum of Comparative Zoology and Department of Organismic and Evolutionary Biology, Harvard University, Cambridge, MA, USA. ✉e-mail: sp587@cam.ac.uk

innervation of frontalmost appendages in radiodonts[22] and of deutocerebral innervation in megacheirans[21]. Doubts have arisen because of the morphological similarities between radiodont appendages and the anterior raptorial appendages of some fossil deuteropods (upper stem-group + crown-group euarthropods), for example *Kylinxia*, isoxyids, and megacheirans[22,29,30], and the apparently abrupt reduction and fusion of radiodont-like protocerebral appendages into a labrum with simultaneous origination of deutocerebral appendages[21]–although this could be explained by the expansion and subdivision of a unipartite brain into three distinct segments[23]. However, these alternative hypotheses require palaeoneurological data from Cambrian euarthropods[2,3,9,26,31,32] and the presence of a labrum in megacheirans[7] to either be substantially reinterpreted[21] or discarded altogether[22].

Here we describe two specimens with a broadly opabiniid-like morphology from the Middle Ordovician (ca. 462 Ma) Castle Bank fauna (Builth Inlier, Wales, UK), a new Burgess Shale-type fauna from the *Didymograptus murchisoni* Biozone[33,34]. These specimens are more than 40 million years younger than all previously known opabiniids[35,36]. Phylogenetic analyses treating shared features of the Castle Bank specimens and radiodonts as homologous support a position for the new specimens crownwards of opabiniids, sister to radiodonts and deuteropods, in the euarthropod stem lineage. Importantly, this paraphyletic association of lower stem-group euarthropods with proboscises indicates that the deuteropod labrum may represent the reduction of an already-fused proboscis, rather than a reduced pair of arthropodized appendages. If instead some shared features between the Castle Bank specimens and radiodonts are considered convergent, phylogenetic analyses retrieve these new specimens as opabiniids. Thus for this latter interpretation the geographic and temporal range of Opabiniidae is extended to a new palaeocontinent (Avalonia) and time period (Ordovician).

## Results
### Geological setting
The studied specimens were collected from Castle Bank, a Burgess Shale-type Konservat-Lagerstätte, located in mid Wales. Initial publications have focused on the sponge fauna[33,34,37], but recently ongoing excavations have revealed a more diverse fauna with members of numerous phyla. Pertinent to this study, other total group euarthropod specimens preserve appendages, carapaces, eyes and other soft internal tissues including guts (SP, JPB, LAM personal observations).

Material was recovered from a small domestic quarry on private land, in the vicinity of Llandrindod Wells (exact location deposited with the specimens) (Supplementary Fig. 1). The quarry lies within the upper part of the Gilwern Volcanic Formation of the Builth-Llandrindod inlier[33]. During the Ordovician the area was part of the Avalonian microcontinent, in the temperate zone of the southern hemisphere. Abundant planktic graptolites date the quarry to the *Didymograptus murchisoni* Biozone (Darriwilian, Middle Ordovician). Lithostratigraphy places the fossils in this study to the upper part of that biozone (c. 462 Ma).

### Systematic palaeontology
Superphylum PANARTHROPODA Nielsen, 1995[38]
 Genus *Mieridduryn* nov.

**Etymology.** From Welsh *mieri* (bramble) and *duryn* (proboscis, snout), meaning "bramble-snout". The dd is pronounced as a soft th, and results from mutation following a feminine noun. Gender *f*.

**Diagnosis.** Panarthropod with head region bearing dorsal sclerite, annulated proboscis with spiniform dorsal projections and radial mouthparts composed of small, sclerotized plates; gut trace leading to posterior-facing mouth; trunk bears large subrectangular dorsolateral flaps with rounded distal margins; dorsolateral flaps bear setal structures on surface facing body midline; annulated lobopods display triangular outline and possess short triangular spines on posterior margin.

*Mieridduryn bonniae* nov. gen. et sp.
 Figures 1–5, Supplementary Figs. 2–4

**Etymology.** After Bonnie Douel, niece of the site owners and fossil devotee; the family has followed and supported the research extensively since the discovery of the fauna.

**Holotype.** NMW.2021.3 G.7 known from part and counterpart. Counterpart preserves anterior portion only.

**Locality and horizon.** Collected from the Darriwilian (Middle Ordovician, *Didymograptus murchisoni* Biozone) Gilwern Volcanic Formation at Castle Bank, near Llandrindod, Powys (UK)[33,34].

**Diagnosis.** As for genus, by monotypy.

**Description.** NMW.2021.3 G.7 preserves the head region and anterior portion of the trunk (Fig. 1). The specimen, which measures -13 mm along the dorsal margin (not including proboscis), is twisted so that the anterior provides an oblique-lateral view, which becomes more oblique-ventral towards the posterior.

The head region preserves evidence for a dorsal sclerite, annulated proboscis, and posterior-facing mouth composed of sclerotized plates. The dorsal sclerite (length -2 mm) has a rounded lateral margin. The proboscis, which is a single structure and not an overlapping pair of appendages, measures ca. 3 mm along its dorsal margin, is annulated, curves ventrally, and displays slender spines at regular intervals (-0.2 mm spacing, one per two to four annulations) along its dorsal margin (Figs. 2a, 3, 4d, Supplementary Fig. 2). Evidence for a posterior-facing mouth is provided by the gut trace, which twists ventrally where it connects to the mouth (Figs. 2a, 4c, Supplementary Fig. 3). The mouth is -0.4 mm in diameter and preserves evidence for small, lightly sclerotized plates subequal in size (Supplementary Fig. 4). Plate boundaries are most visible at the anterior and posterior of the ring; the left and right sides display greater disarticulation and are less complete (Supplementary Fig. 4).

The trunk bears two sets of appendages: dorsolateral flaps and ventral lobopodous limbs. Flaps intersect with faint curved boundaries, interpreted as the body margin, and display a subrectangular outline with rounded margins (Fig. 5b). Flaps decrease in size towards the posterior (lf1 measures -3 mm along its long axis, lf3 -2 mm). Although most flaps display a smooth external surface, the anterior-most flaps provide evidence for internal linear features, interpreted as strengthening rays (Figs. 2b, c, 4c). These are visible in the left anterior flap, which has been split obliquely, revealing interior structures. The strengthening rays run parallel to the long axis of the flap and cover most of the flap width. Both anterior flaps display a darker region that covers most of the surface facing the body midline. This darker region preserves fine lines that run perpendicular to the strengthening rays, interpreted as setal structures. Additional setal structures protrude from underneath the posterior margins of other flaps (e.g. Fig. 5b). Towards the posterior, the body is twisted and the swimming flaps become preserved more obliquely. This reveals a second ventral series of lobopodous limbs (Fig. 5c, d). These limbs are triangular in outline and display lineations that run perpendicular to the long axis, interpreted as annuli. Spines protrude from the posterior margin (Fig. 5c, d).

**Remarks.** The unique combination of characters, including some previously considered exclusive to opabiniids (annulated proboscis) and radiodonts (dorsal spines on the protocerebral appendage)

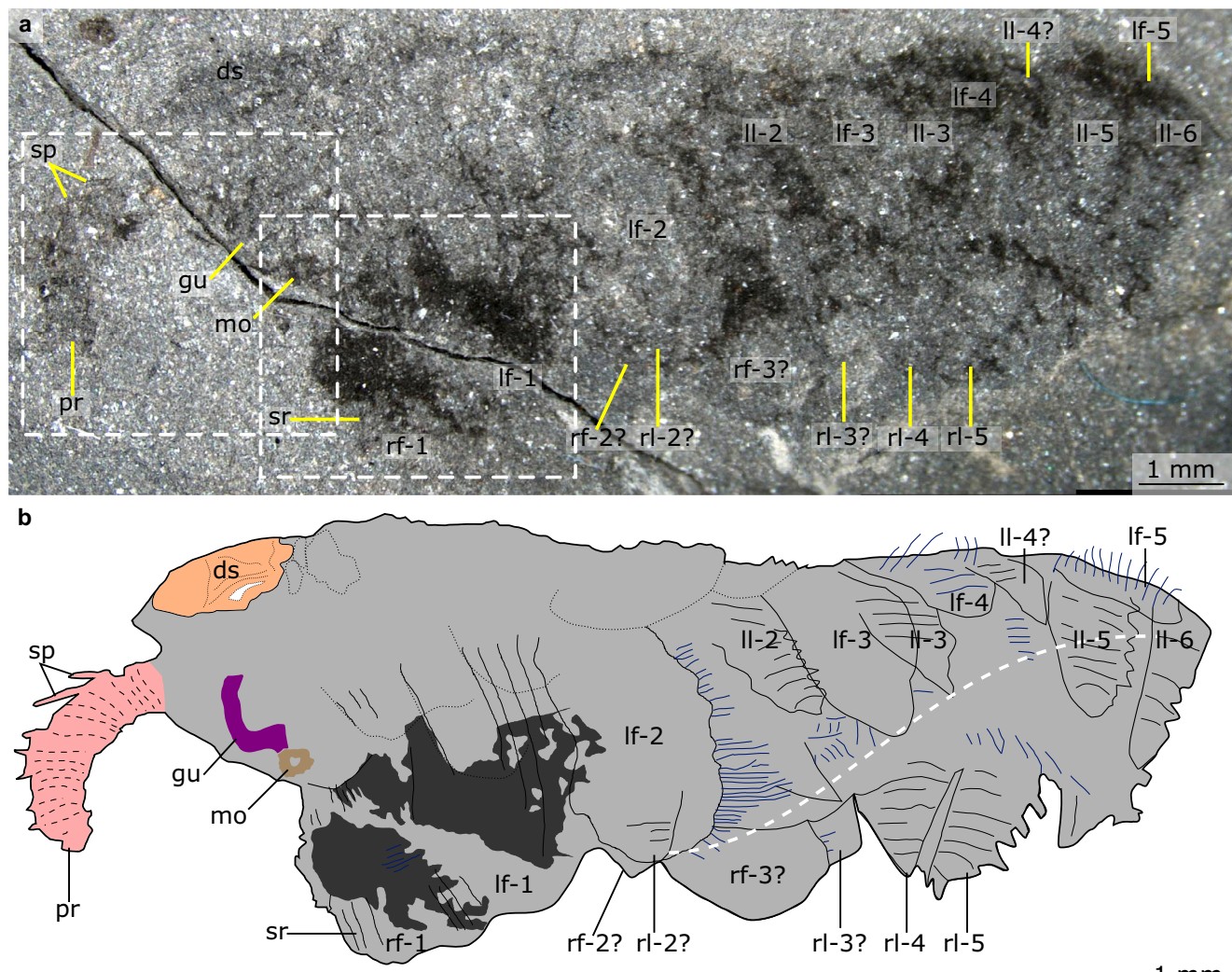

**Fig. 1 | *Mieridduryn bonniae* nov. gen. et sp. from the Castle Bank Biota (NMW.2021.3 G.7). a** Overview of whole specimen. Boxes indicate areas depicted in Fig. 2. **b** Explanatory drawing of (**a**). Dashed white line indicates ventralmost point of left dorsolateral flaps, and demonstrates the twisted nature of the specimen. Blue lines indicate filamentous setal structures. ds dorsal sclerite, gu gut, lf left dorsolateral flap, ll left lobopod, mo mouth, pr proboscis, rf right dorsolateral flap, rl right lobopod, sr strengthening ray, sp dorsal spines on proboscis.

supports the erection of a new genus and species. Phylogenetic analyses support *Mieridduryn bonniae* nov. gen. et sp. as the most stemward member of the euarthropod stem lineage that bears dorsal spines on the protocerebral appendage and dorsolateral flaps with strengthening rays, and most crownward member to exhibit lobopodous ventral limbs (further remarks in Supplementary Information).

Castle Bank euarthropod A

Figures 6–9, Supplementary Fig. 5

**Material, locality, and horizon.** NMW.2021.3 G.8, known from part and counterpart. Collected from the Darriwilian (Middle Ordovician, *Didymograptus murchisoni* Biozone) Gilwern Volcanic Formation at Castle Bank, near Llandrindod, Powys (UK)[33,34].

**Description.** NMW.2021.3 G.8 is a complete specimen preserved compressed to give a lateral view and measures ~3 mm along the dorsal margin (Fig. 6).

The head region preserves evidence for what appear to be two lateral cephalic sclerites proximal to an annulated proboscis (Figs. 6, 7). A pair of trapezoidal sclerites <1 mm in length, which are compressed and superimposed on each other, have been rotated forwards to cover the anterior of the head and proximal part of the proboscis (Supplementary Fig. 5). The preserved portion of the proboscis is less than 1 mm in length. The cephalic sclerites preserve a marginal rim, and the element covering the right side of the head also displays numerous triangular spines on the anterior and ventral margins (Fig. 7a, Supplementary Fig. 5). The proboscis, which curves ventrally and is likely incomplete distally, displays annulations and a dark linear feature that runs parallel to the long axis, interpreted as an internal canal, as well as regularly spaced (approximately one per three to four annulations) stout triangular dorsal spines (Figs. 7, 8c). The proboscis is more similar in preservation to the trapezoidal sclerites than to the trunk.

The trunk, which measures ca. 2 mm along the dorsal margin, is curved, narrow and tapers slightly to the posterior. The trunk displays a wrinkled texture. At the anterior (just behind the lateral carapace elements), faint subrectangular elements sit dorsal to the body region (Fig. 7a; Supplementary Fig. 5). Further to the posterior, indents in the dorsal margin and faint transverse lineations indicate the position of intersegmental furrows and associated subrectangular lateral swimming flaps (Figs. 6, 9). Fainter lines overlain by wrinkles preserved towards the middle of the trunk are also tentatively identified as dorsal furrows and lateral flaps, and indicate the presence of at least 12 furrows and flaps (Fig. 6).

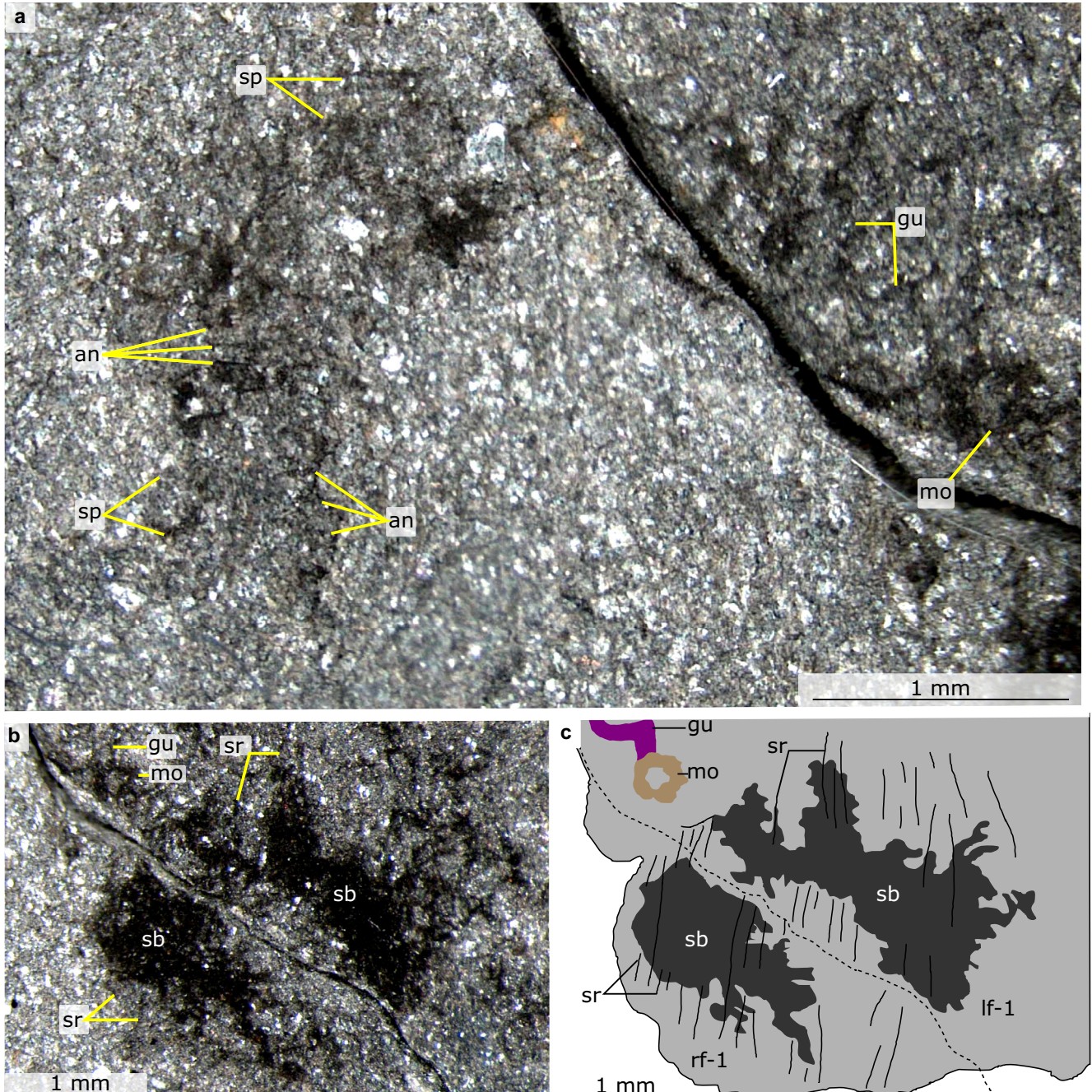

**Fig. 2 | Details of anterior of *Mieridduryn bonniae* nov. gen. et sp. (NMW.2021.3 G.7). a** Anterior of the head region including proboscis, mouth, gut, view of anterior details including spinous proboscis, mouth and gut. **b** Mouth and anterior flaps with strengthening rays. **c** Explanatory drawing of (**b**). an annulations on proboscis, gu gut, lf left dorsolateral flap, mo mouth, rf right dorsolateral flap, sb setal blades, sp spines on proboscis, sr strengthening rays.

The body terminates in a caudal fan -0.25 mm long, composed of triangular caudal blades with spinose lateral and posterior margins. Three blades that widen to the posterior can be observed on the near (right) side, whereas two are observed on the left side (Figs. 8b, 9).

**Remarks.** The wrinkled texture of the body, change in slope on the dorsal margin and orientation of the paired head sclerites (rotated forwards), as well as the lack of internal structures (e.g. guts, nervous system) suggest that this specimen may represent a moult rather than a carcass[39], though this is not conclusive. The presence of an internal cavity does not refute this, as similar structures have been reported from isolated radiodont appendages (e.g. ref. 15 Fig. 13). If this specimen does represent a moult, then it is possible that it bore only a single cephalic sclerite. The appearance of a paired sclerite may have been caused by folding, shearing and breakage of a single, larger, sclerite during the moulting process. The smaller 'rectangular elements' posterior to this apparently paired sclerite may represent additional broken fragments.

If the presence of multiple head sclerites in NMW.2021.3 G.8 (rather than one in NMW.2021.3 G.7) is confirmed, then this difference and the distinct morphology of the dorsal spines on the proboscis together suggest that NMW.2021.3 G.7 may represent a distinct species. Other potentially diagnostic features of NMW.2021.3 G.8, such as the spinose caudal fan, are not preserved in NMW.2021.3 G.7, but are present in opabiniids *Opabinia* and *Utaurora*[36].

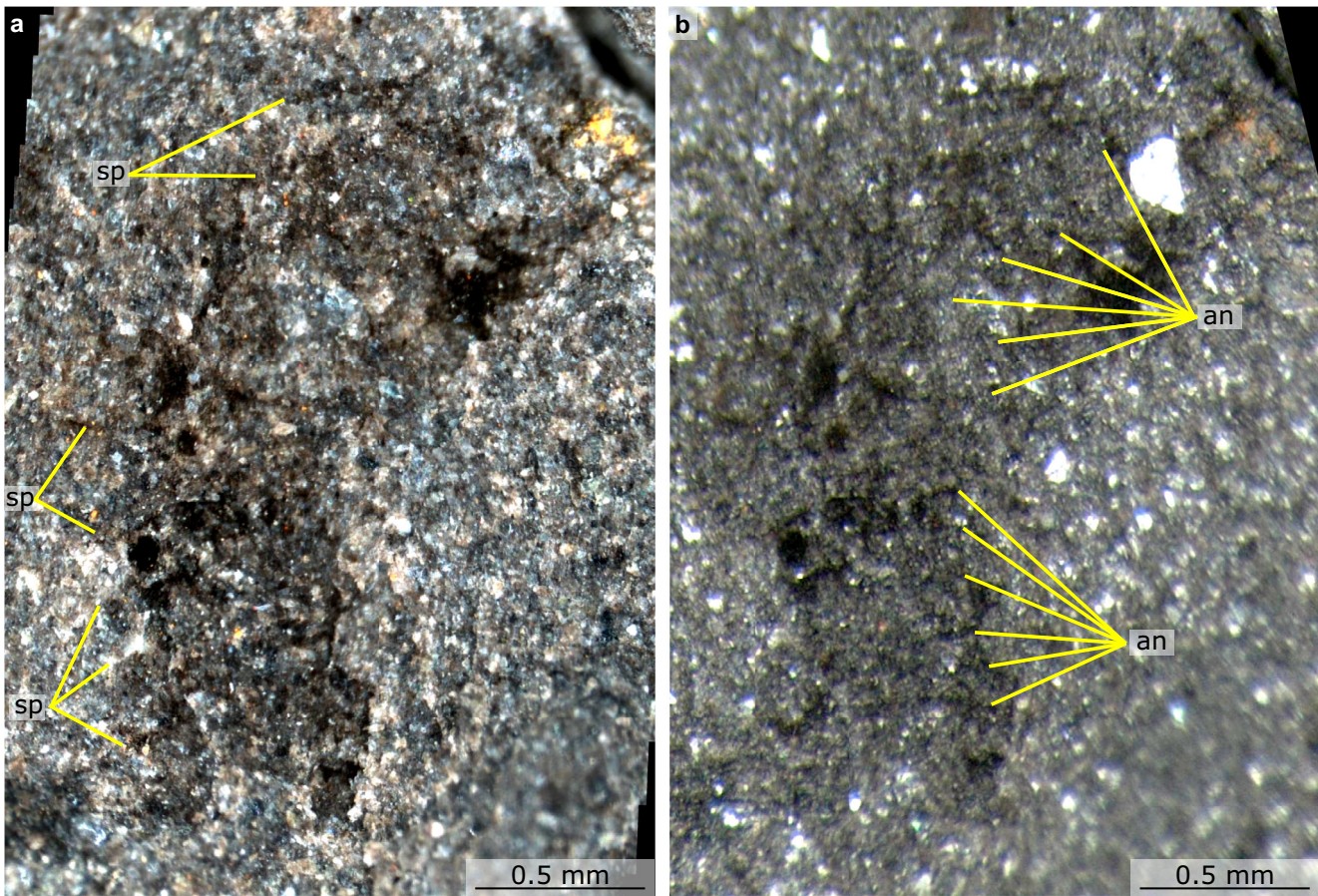

**Fig. 3 | Details of proboscis of *Mieridduryn bonniae* nov. gen. et sp. (NMW.2021.3 G.7) under different lighting conditions. a** S8 microscope, cross-polarised light, stitched images, with contrast increased. **b** M125 microscope under high angle light. an annulation, sp spine.

Alternatively, NMW.2021.3 G.8 could also be regarded as an earlier ontogenetic stage of *Mieridduryn bonniae* than NMW.2021.3 G.7. At ~3 mm, euarthropod A falls within the size range of some larger crustacean larvae (e.g. decapods, remipedes)[40]; thus, postembryonic morphological changes such as differing spine form on the proboscis and number of head sclerites could explain the observed differences in the two specimens. Within radiodonts the number of head sclerites is stable at the family level, and does not change during ontogeny (in at least one species, *Lyrarapax unguispinus*[41]), but the number of carapace elements does change during ontogeny in other euarthropods. For example, in the fossil phosphatocopine *Dabashanella* and the living ostracod *Manawa*, a univalved carapace develops into a bivalved one[42,43]. Metamorphosis is expected to be ancestral for panarthropods[44] and has been observed in some members of the upper stem group (e.g. the megacheiran *Leanchoilia*[45], but not all, see ref. [46]); hence a comparatively minor morphological change during ontogeny for NMW.2021.3 G.7 and NMW.2021.3 G.8 cannot be ruled out with the current data. Thus, we leave the smaller specimen in open nomenclature, and consider both possibilities (a distinct species or an earlier ontogenetic stage of *Mieridduryn* nov. gen.) for this specimen in our phylogenetic analyses (section below).

Further discussion of the relationship of *Mieridduryn* nov. gen. and euarthropod A to opabiniids is provided in Supplementary Information.

## Phylogenetic results

In our phylogenetic analyses, we considered two possibilities concerning the new specimens−that they represented two ontogenetic stages of one species, or two distinct taxa. We also considered whether the inclusion of controversial deuteropod taxa would change the topology of taxa stemwards of radiodonts. Broad agreement in the results considering one or two Castle Bank terminals, and sensitivity analyses including additional deuteropod taxa, consistently support a topology of a paraphyletic grade of lower stem-group euarthropods with fused protocerebral appendages (Supplementary Discussion).

The consensus Bayesian Inference (BI) and Maximum Parsimony (MP) topologies resolve the Castle Bank taxa in the lower stem group of Euarthropoda, more closely related to radiodonts and deuteropods than to opabiniids (Fig. 10, Supplementary Figs. 6–9). Notably, all MP trees support radiodonts as sister group to deuteropods, unlike previous analyses using similar datasets[6,36,47].

We visualised the tree topologies in multidimensional (MDS) treespace[36]−where each point denotes a tree in the posterior sample (BI) or most parsimonious tree set (MP), and the distance between the points is lower for more similar trees (Supplementary Figs. 8, 9). When two Castle Bank terminals were considered, there is near-complete overlap in the areas occupied by trees resolving a monophyletic group of proboscis-bearing stem-group euarthropods (Castle Bank specimens + *Opabinia* + *Utaurora*) and those resolving a paraphyletic grade (Castle Bank taxa sister to Radiodonta + Deuteropoda) (Supplementary Fig. 8). This overlap extends through the first three MDS axes, and thus the position of the Castle Bank taxa in the phylogeny is not the main driver of variation within the results. When one Castle Bank terminal is considered, slight separation in the two groups of trees can be observed (Supplementary Fig. 9).

Examination of the phylogenetic support for analyses considering one or two Castle Bank species reveals that a small majority of BI (52% and 69%, respectively) and all MP analyses support placing the Castle

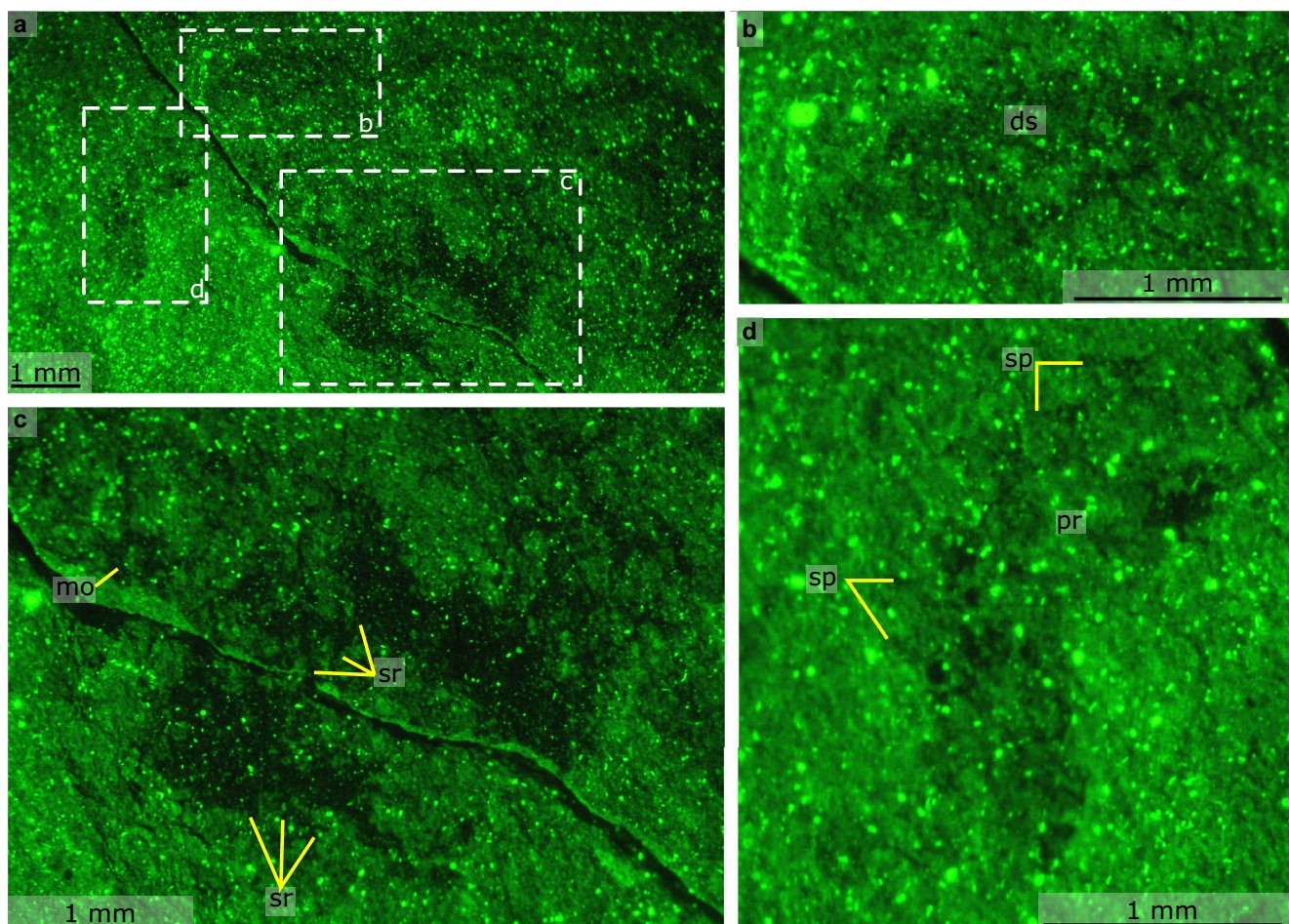

**Fig. 4 | Details of anterior of *Mieridduryn bonniae* nov. gen. et sp. (NMW.2021.3 G.7) photographed using fluorescence. a** Anterior of the head region including proboscis, mouth, and gut. **b** Dorsal sclerite. **c** Mouth and anterior flaps with strengthening rays. **d** Proboscis with spines. ds dorsal sclerite, pr proboscis, sp spine, sr strengthening rays.

Bank specimens in a monophyletic group with Radiodonta and Deuteropoda (Supplementary Tables 1, 2; Supplementary Figs. 6, 7). The best-supported alternative—that the Castle Bank specimens are opabiniids—is supported by 32% of trees when the two specimens are considered to represent one species, and 13% of trees when the two specimens are treated separately (Supplementary Tables 1, 2). For both sets of analyses, a larger proportion of trees supports a paraphyletic proboscis grade (59% for one species and 76% for two species) than a monophyletic proboscis group (33% for one species and 14% for two species) (Supplementary Tables 1, 2). The recovered paraphyletic grade of proboscis-bearing stem-group euarthropods was robust to the inclusion of additional taxa recently considered sister to all other deuteropods (*Kylinxia* and *Parapeytoia*)[21,30] and multiple coding strategies relating to the segmental affinities of *Parapeytoia* frontalmost appendages (Supplementary Figs. 10, 11, Supplementary Discussion).

We conducted further analyses that considered particular morphological features with uncertain homology observed in the Castle Bank specimens to be convergent with those observed in radiodonts and other total group euarthropods. We explored the role of carapace elements, dorsal spines on the proboscis, and internal strengthening rays, both individually and in combination, in recovering a paraphyletic grade of proboscis-bearing stem-group euarthropods. For these analyses, we considered the Castle Bank specimens to comprise a single species. When any, or all three, of these features were considered to have evolved convergently in the Castle Bank specimens

and radiodonts, *Mieridduryn* was no longer recovered crownwards of opabiniids. For three of the four analyses a monophyletic Opabiniidae consisting of *Opabinia*, *Utaurora* and *Mieridduryn* was recovered, whereas if only the strengthening rays were considered convergent, a polytomy of *Opabinia*, *Mieridduryn*, *Utaurora* and a clade comprising radiodonts and deuteropods was recovered (Supplementary Fig. 12). In all these sensitivity analyses, more trees in the posterior sample retrieved the Castle Bank terminal in a monophyletic Opabiniidae than in a paraphyletic grade of proboscis-bearing stem-group euarthropods (Supplementary Table 3).

Visualisation of these results in multidimensional treespace recovered two overlapping islands, one supporting a monophyletic group of proboscis-bearing stem-group euarthropods, the other a paraphyletic grade (Supplementary Fig. 13). The overlapping region is sampled by all five analyses, however there are differences in the location of trees in the posterior samples across all the analyses. When no characters are considered convergent, more of the posterior sample falls within the positive part of MDS axis 1, reflecting the presence of more trees recovering a paraphyletic grade of proboscis-bearing euarthropods. All posterior samples in analyses in which at least one character is regarded as convergent explore more of the negative part of MDS axis 1, reflecting a greater number of trees that recover a monophyletic grouping of proboscis-bearing euarthropods (Supplementary Fig. 13). When all three characters are considered convergent, the treespace occupied is most negative on MDS axis 1 of all five analyses, reflecting the presence of the highest

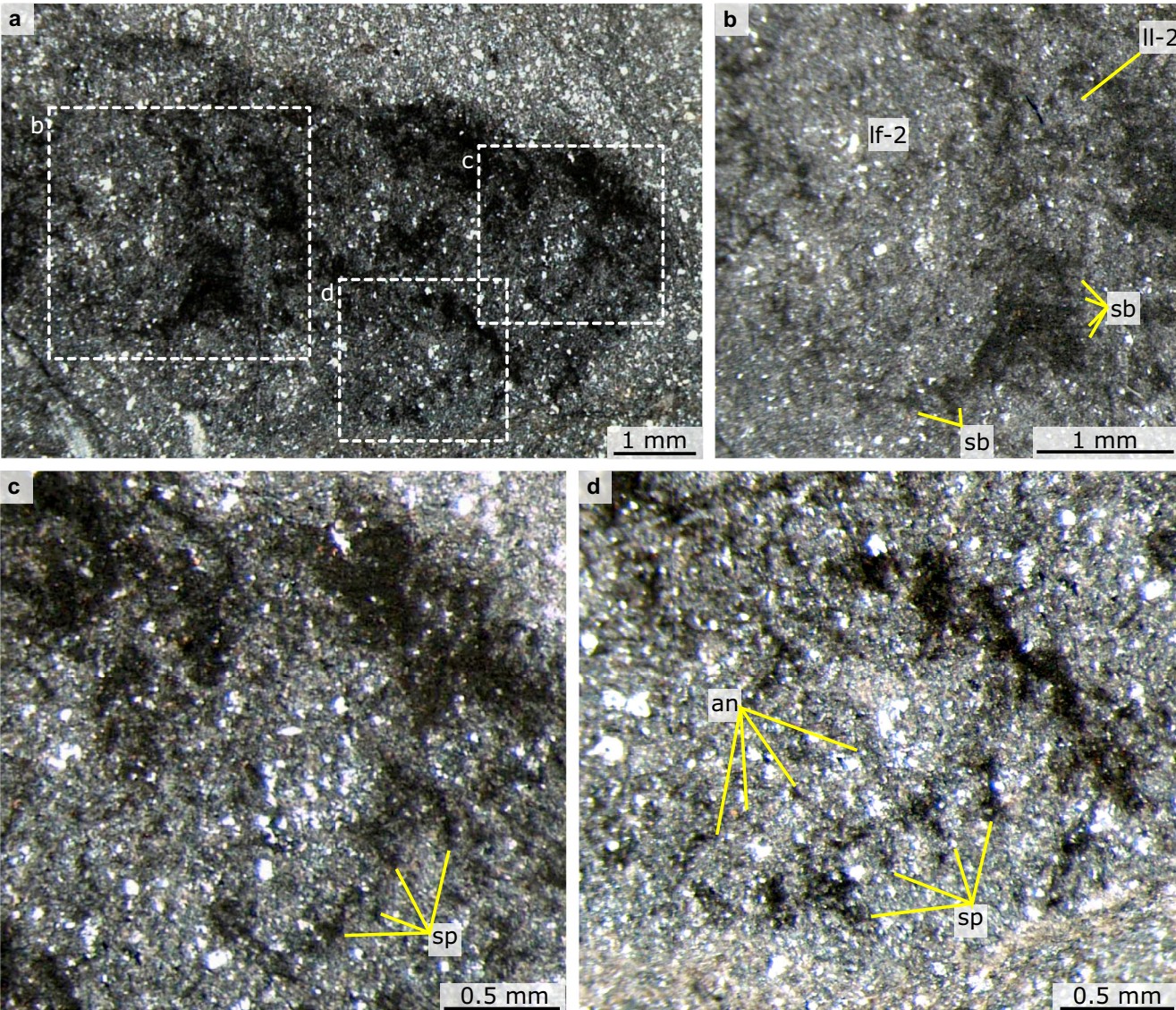

**Fig. 5 | Details of trunk of *Mieridduryn bonniae* nov. gen. et sp. (NMW.2021.3 G.7). a** Overview of trunk. **b** Details of left dorsolateral flap and associated setal blades. **c, d** details of lobopodous limbs with annulations and spines. an annulations on lobopodous limbs, lf left dorsolateral flap, ll left lobopodous limb, sb setal blade, sp spines on posterior margin of lobopodous limb.

proportion of monophyletic proboscis trees in the posterior sample of all analyses visualised (Supplementary Fig. 13, Supplementary Table 3).

## Discussion

The presence of a slender body with dorsal furrows, dorsolateral flaps plus lobopodous limbs, anterior proboscis, and spinose caudal blades in the Castle Bank specimens demonstrates that an opabiniid morphology persisted until at least the Middle Ordovician, more than 40 million years longer than previously known[35,36]. Indeed, until recently, *Opabinia* was the only reported stem-group euarthropod with a proboscis[35,36]. The suite of phylogenetic analyses demonstrate that, if features shared between the Castle Bank specimens and radiodonts may be considered homologous, the Castle Bank specimens are the latest diverging members of a paraphyletic grade of stem-group euarthropods with proboscises. If instead at least one of these characters is considered to have evolved convergently in the Castle Bank specimens and in radiodonts, then the new specimens are best considered as opabiniids (in a monophyletic group with *Opabinia* and *Utaurora*). Both scenarios are congruous with developmental data, but

have different implications for our understanding of the evolution of the group.

Importantly these interpretations treat the proboscis as a fused pair of protocerebral appendages e.g. refs. 48–50 rather than considering it a novel structure. Multiple lines of evidence support this interpretation. Firstly, while no palaeoneurological tissues have been reported from an opabiniid, neural tissues are known from taxa which phylogenetically bracket opabiniids and the Castle Bank specimens: *Kerygmachela* and the radiodont *Lyrarapax*[3,25], which support a protocerebral affinity (note that fossilised neural tissues in a second radiodont taxon, *Stanleycaris*, argued to support a deutocerebral affinity for radiodont appendages[51] can also be interpreted as innervated by the protocerebrum[52]). Secondly all members of the euarthropod lower stem group are widely thought to have a head composed of a single segment that bears one pair of appendages[11,14] (though see Ref. 51 for an alternative view that is itself open to reinterpretation[52]). In opabiniids and the Castle Bank specimens, only a single proboscis is present in the head region, and no other appendage-like structures. Either these animals lost paired protocerebral appendages and developed a novel proboscis, or the

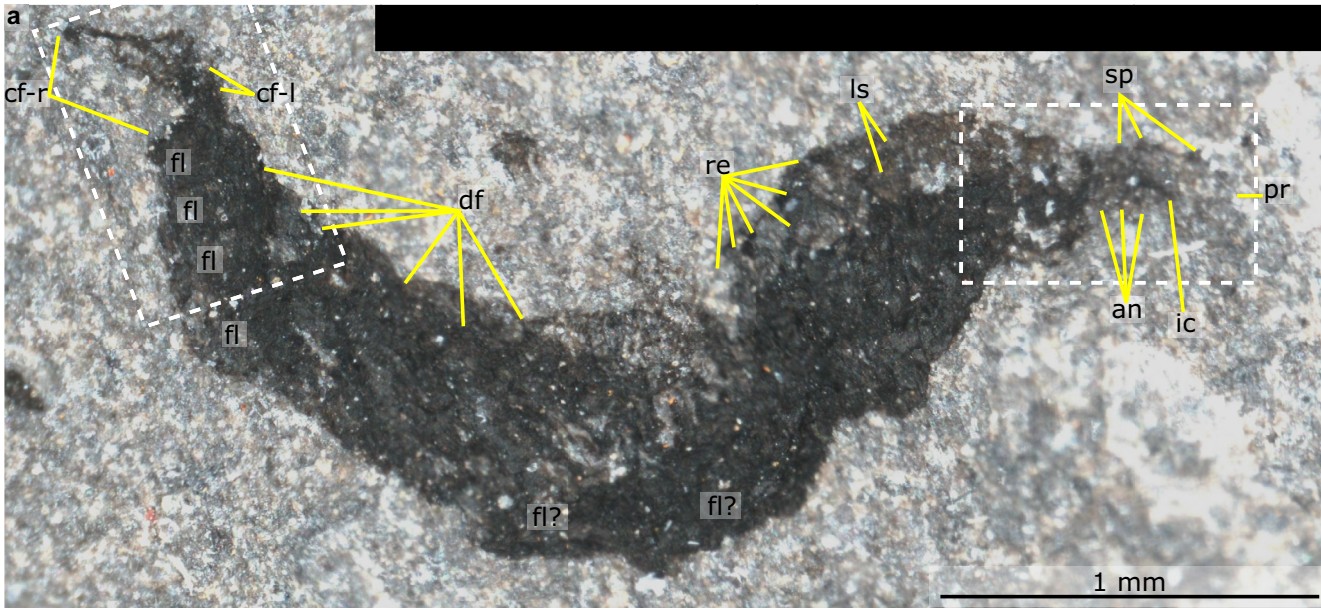

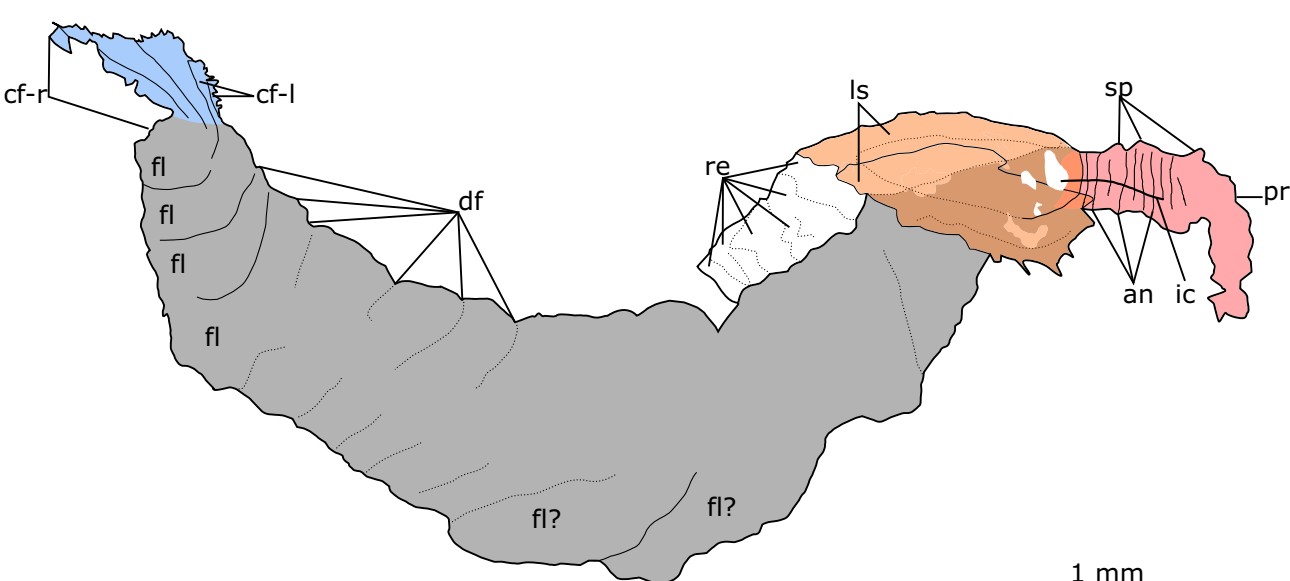

**Fig. 6 | Castle Bank euarthropod A (NMW.2021.3 G.8) from the Castle Bank biota. a** Overview of whole specimen. Boxes indicate areas detailed in Fig. 7. **b** Explanatory drawing of (**a**). an annulations of proboscis, cf-l left blades of caudal fan, cf-r right blades of caudal fan, df dorsal furrow in trunk, fl dorsolateral flap, ic internal canal of proboscis, ls lateral sclerite, pr proboscis, re subrectangular elements posterior to lateral sclerites, sp spine on proboscis.

proboscis represents a fused pair of appendages. Thirdly, the proboscis shares characters with frontal appendages of gilled lobopodians (annulations) and radiodonts (endites in the distal claw of *Opabinia* and dorsal spines in the Castle Bank specimens). If our assumption is incorrect, and the proboscis of opabiniids and Castle Bank specimens represents a novel feature not present in the remainder of the euarthropod stem lineage, then it would have limited implications for the evolution of the head in the remainder of the group, and similarities with frontal appendages of radiodonts such as dorsal spines would be considered convergent.

### Homologous characters, paraphyletic grade of proboscis-bearing lower stem-group euarthropods
The reconstructed paraphyletic grade of stem-group euarthropods with proboscises (opabiniids and the Castle Bank specimens) suggests

that a fused proboscis may have been present in the last common ancestor of opabiniids and deuteropods, rather than being exclusive to opabiniids (Fig. 10).

These topologies suggest that the deuteropod labrum, rather than having evolved from a reduced pair of arthropodized appendages, may instead represent a reduced, sclerotized, already fused, proboscis. Euarthropod trunk appendages and labrum were both likely modified from an ancestral appendage patterning network[53]. Embryological evidence demonstrates that the labrum develops from the fusion of paired anterior structures[54], as is also inferred for the proboscis of *Opabinia*[55], which is congruent with a homology to the protocerebral appendages of lower stem-group euarthropods[14,53]. Genes that regulate the upstream components of the proximodistal axis of trunk appendages are also expressed in labrum development, adding further support for an appendicular origin of the labrum[53,56]. However,

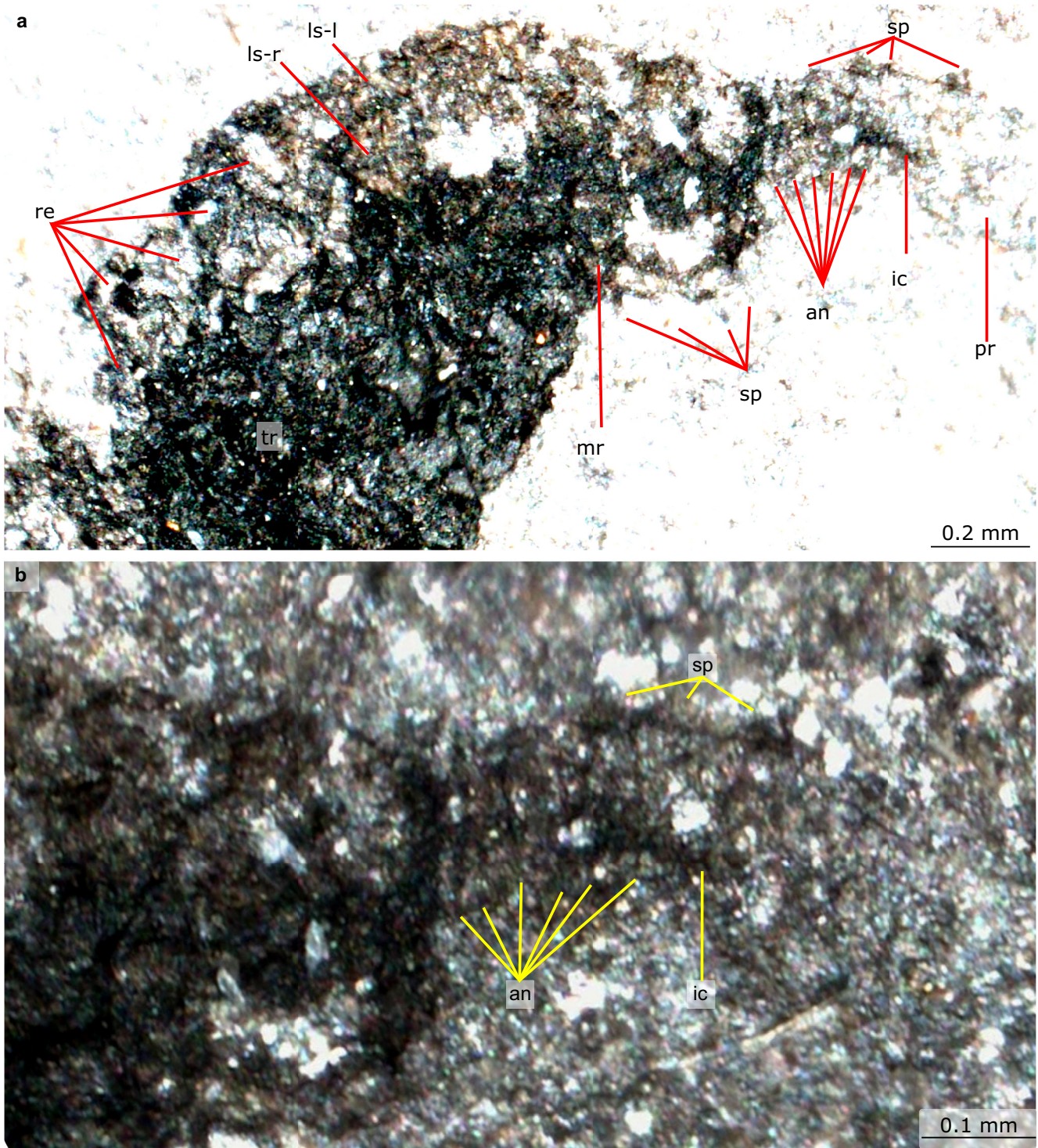

**Fig. 7 | Details of anterior of Castle Bank euarthropod A (NMW.2021.3 G.8).**
**a** Anterior of head including lateral sclarites, rectangular elements and spinose proboscis with annulations (**b**) Details of spinose proboscis and annulations; an annulation, ic internal canal of proboscis, ls-l left lateral sclerite, ls-r right lateral sclerite, mr marginal rim to carapace element, pr proboscis, re rectangular element posterior to lateral sclerites, sp spine.

downstream components of the trunk appendage patterning network, including most genes involved in joint and segment formation, are not active during labrum development[53,56]. This evidence raises the intriguing possibility that rather than representing a segmented appendage that had lost joints and the corresponding joint-forming developmental framework[53], the labrum instead may have originated from a proboscis with proximo-distal differentiation, but no joints or associated joint-formation gene regulation. As it is not possible to

prove or disprove a loss, a slightly less parsimonious alternative remains plausible. In this alternative scenario, joint-formation originated in the protocerebral appendages of the common ancestor of radiodonts and deuteropods and remained in the protocerebral appendages only in the lineage leading to radiodonts. In deuteropods, genes involved in joint development would have been co-opted into the deutocerebral and subsequent appendages, and lost in the labrum[53,55], perhaps associated with an expansion and subdivision of

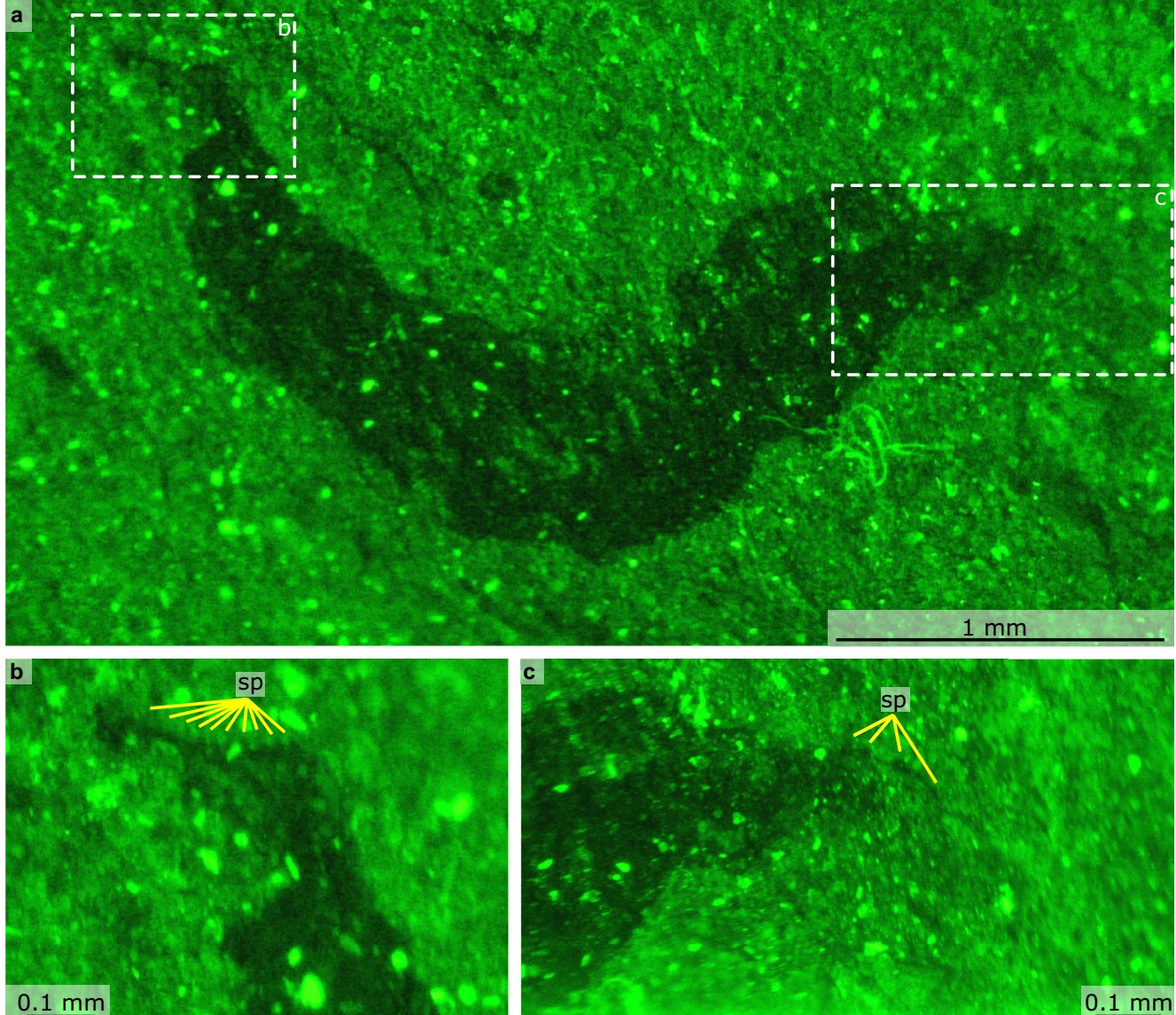

**Fig. 8 | Details of Castle Bank euarthropod A (NMW.2021.3 G.8), photographed using fluorescence. a** Overview of whole specimen. **b** Details of tail fan with spinose margin. **c** Details of proboscis with dorsal spines. sp spine.

the unipartite head into three distinct segments[23]. Regardless, the association of a posterior-facing mouth with a fused protocerebral appendage appears to have originated in the lower stem group of Euarthropoda (evinced by *Opabinia*), a deeper root than previously appreciated (Fig. 10).

Sclerotization is one of three key features of arthropodized appendages[57] and a prerequisite for the other two—segmentation and articulation. Indeed, comparison between the proboscises of *Opabinia* and the Castle Bank specimens indicates that the latter were more sclerotized than the former. The proboscis of *Opabinia* is preserved like the trunk, as a dark carbonaceous film[35,58–60], whereas the Castle Bank specimens bear spines on the dorsal margin of the proboscis, and preservation of this morphological feature resembles the lightly sclerotized carapace elements more than the body (Figs. 6, 7). This evidence suggests increasing sclerotization of the proboscis through the paraphyletic lineage of proboscis-bearing stem-group euarthropods. These results also suggest that the protocerebral appendage became secondarily divided in radiodonts. For these topologies, a single fusion of the protocerebral appendage in opabiniids and subsequent separation in radiodonts is slightly more parsimonious than

the alternative: multiple independent fusions of the protocerebral appendage in opabiniids, the Castle Bank specimens, and deuteropods (Fig. 10). This topology also requires that arthropodized appendages arose convergently in radiodonts and deuteropods. The regularly spaced dorsal spines may have provided a repeated pattern along the proximodistal axis that might have been co-opted during development for joint formation[53] in the frontal appendages of radiodonts (Supplementary Discussion). In deuteropods, this same repeated pattern may have been co-opted by the deutocerebral appendage following the expansion of the head into multiple segments and subfunctionalization of head appendages[23].

A paraphyletic grade of proboscis-bearing stem-group euarthropods offers a possible solution to an outstanding issue in the evolution of the protocerebral appendage within Euarthropoda: the apparently simultaneous appearance of a labrum and arthropodized deutocerebral appendages in the fossil record[21]. This topology facilitates functional continuity, as it requires a transition between a hypothetical ancestral animal with a protocerebral proboscis (and presumably no deutocerebrum) and a deuteropod with labrum and specialised deutocerebral appendages. A prehensile proboscis is

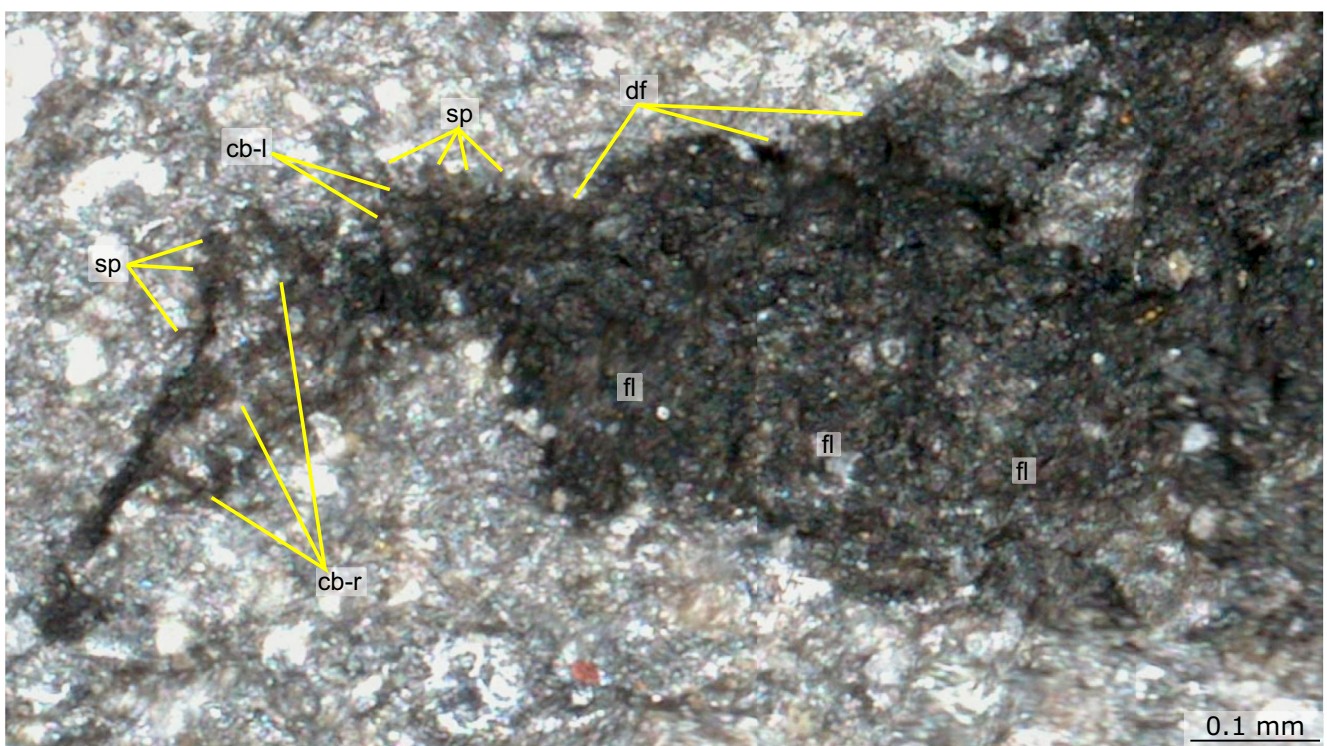

**Fig. 9 | Details of posterior body and tail fan of Castle Bank euarthropod A (NMW.2021.3 G.8).** cb-l left caudal blade, cb-r right caudal blade, df dorsal furrow, fl lateral flap, sp spine.

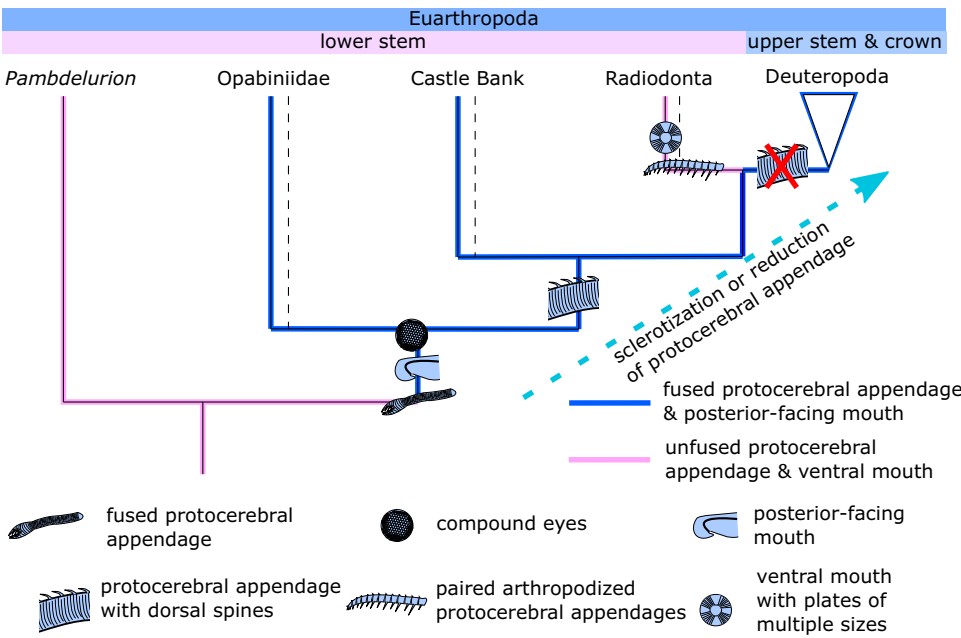

**Fig. 10 | Simplified results of phylogenetic analyses and a putative pathway of evolution of key head characters within the euarthropod lower stem group.** Dotted lines indicate nodes where some analyses resolved polyphyletic or paraphyletic relationships. Full phylogenetic results provided in Supplementary Figs. 6–11.

capable of both food capture and transport to the mouth, but is more morphologically constrained than paired appendages. This morphological flexibility of paired appendages is supported by the far greater diversity of radiodonts compared to opabiniids, and the array of ecological niches facilitated by the disparity of frontal appendages in the former[1,15,16,18,19,41,59,61–63]. Thus, the evolution of specialised food-

capturing deutocerebral appendages in deuteropods could have augmented the feeding capabilities of a proboscis-bearing animal, which in turn could have facilitated the takeover of food-capture responsibilities by the deutocerebral appendages, with the protocerebral appendage instead specialising for holding food in position. This new specialisation would have allowed reduction of the proboscis and

migration towards the posterior-facing mouth. Thus after subdivision of the head, the appendages of each segment evolved independent specialisations for distinct functions[23], with the genetic basis of both a fused labrum and repeated pattern for joints both potentially present in the proboscis of the Castle Bank specimens.

## Convergent characters, monophyletic opabiniids

The topology recovered above (Fig. 10) assumes that the carapace, dorsal spines on the proboscis, and presence of strengthening rays in the lateral flaps, are homologous in the Castle Bank specimens and radiodonts. If instead some or all of these characters are considered to have evolved convergently in these two groups, an alternative topology is recovered. In this alternative, the Castle Bank specimens are resolved in a monophyletic group with *Opabinia* and *Utaurora*. This alternative scenario, in which the Castle Bank specimens are recovered as opabiniids, also has important implications for our understanding of euarthropod evolution, and is informative about some aspects of the morphology of *Opabinia*.

This topology implies the same broad-scale scenario for the evolution of features in the lower stem group as recovered by previous studies (e.g. ref. 36) but requires that some morphological features that evolved in radiodonts and were retained in upper stem-group euarthropods, such as a dorsal carapace[64] and/or strengthened swimming flaps[17], were convergently evolved in opabiniids during the Ordovician. In addition, the presence of dorsal spines on the protocerebral appendage, currently only known in radiodonts (e.g. ref. 65), may also have convergently evolved in one opabiniid. Importantly, these features were either convergently evolved in two groups, opabiniids and radiodonts, within the lower stem group of Euarthropoda, or they first evolved in the common ancestor of opabiniids, radiodonts, and deuteropods, rather than the common ancestor of only radiodonts and deuteropods as previously thought. Our interpretation that the proboscis represents a fused pair of appendages homologous to the frontal appendages of radiodonts is likely to have less influence on the outcome of these analyses, as a proboscis is only found in a monophyletic group.

Treating the Castle Bank specimens as opabiniids also sheds light on some controversial aspects of the morphology of *Opabinia*. Dark triangular regions in *Opabinia* have been interpreted as either lobopods[66] or as extensions of the gut[60]. Comparison with *Mieridduryn* supports the former interpretation. The differing locations and morphology of setal structures in *Mieridduryn* and *Utaurora* (dorsal surface of the body and anterior margin of flaps)[36] indicates that opabiniids displayed variability in the location and morphology of their setal structures, comparable to the range of morphology in radiodonts[1,18,19,36], even if the exact position of setal structures in *Opabinia* remains debated[60,66].

More broadly the analyses which recover *Mieridduryn* as an opabiniid would demonstrate an increased morphological disparity and size variation for Opabiniidae, extend the temporal range for this group until at least the Middle Ordovician, over 40 million years longer than previously thought, and expand the palaeogeographic range to a new palaeocontinent: Avalonia. *Opabinia* and *Utaurora* are each only known from single deposits in Laurentia[35,36].

In summary, two new stem-group euarthropod specimens with opabiniid-like morphology from the Middle Ordovician Castle Bank Biota (Wales, UK) are resolved in the euarthropod lower stem group. Phylogenetic analyses either recover these specimens as the outgroup to the clade of radiodonts and deuteropods, or, if some similarities between them and radiodonts are considered the result of evolutionary convergence, as opabiniids. A paraphyletic relationship of opabiniids and the Castle Bank specimens suggests that a fused protocerebral appendage and a posterior-facing mouth, both characters that are typical of deuteropods, were present together in the lower stem group. Under this scenario, the fused proboscis may have been

reduced to become the labrum of deuteropods, with the arthropodized frontal appendages of radiodonts representing an alternative fate for the protocerebral appendage, and suggesting that morphological similarities between radiodont appendages and those of upper stem-group euarthropods are most likely not homologous. If instead the Castle Bank specimens are considered opabiniids, this new discovery greatly extends both the geographic and temporal ranges for the family, increases the morphological disparity of Opabiniidae, and demonstrates convergence in some morphological features in radiodonts and opabiniids.

## Methods

Specimens NMW.2021.3 G.7 and NMW.2021.3 G.8 were collected from a small quarry on private land. The quarry is located in a livestock (sheep grazing) field adjacent to the owners' house. Full permission was granted by the landowners for excavation and deposition of specimens, and the land does not fall under any restrictions requiring permits for the work. The specimens are retained in the country of origin (UK), deposited in the Amgueddfa Cymru−National Museum Wales, Cardiff, UK (NMW).

Material was imaged with Leica S8 APO and M125 stereo-microscopes, combined with a HiChromeAF MET camera and cross-polarised lighting. Fluorescence imaging conducted with Nightsea light source and filter adaptors for Leica S8APO, using Royal Blue (440–460 nm) excitation wavelength. Images were processed using Glimpse Image Editor 0.2.0. Variation in lighting conditions and microscope revealed different morphological details, so composite drawings were used to illustrate specimens. Figures and line drawings were constructed using Inkscape 1.0.

Three fossil taxa (*Mieridduryn bonniae* nov. gen. et sp., NMW.2021.3 G.8, and *Buccaspinea cooperi* Pates et al.[19]) were added to an existing morphological matrix (ref. 36, itself modified from ref. 47), for a total of 57 taxa (11 extant, 46 fossil). *Mieridduryn* nov. gen. was scored a second time, where NMW.2021.3 G.8 was considered an earlier ontogenetic stage, with this matrix comprising 56 taxa (11 extant, 45 fossil). Six characters were added to the original matrix, and two removed, to give a total of 129 characters. Two additional fossil taxa (*Kylinxia zhangi* Zeng et al. 2020[30] and *Parapeytoia yunnanensis* Hou et al. 1995[49]) were coded for sensitivity analyses (more details in Supplementary Discussion). An additional character (character 55) was added to the matrices including *Parapeytoia yunnanensis*, so these analyses include 59 taxa and 130 characters. *Parapeytoia yunnanensis* was scored twice, once with frontalmost appendages considered protocerebral in origin[21], and a second time treated as deutocerebral[17]. We conducted further sensitivity analyses that considered the impact of carapace elements, dorsal spines on the proboscis, and internal strengthening rays, on our topologies. For these analyses we treated the Castle Bank specimens as a single terminal, and added additional characters and character states that allowed these morphological features to be treated as convergently evolved in the Castle Bank specimens, both individually and in combination (131 characters total for all three features treated as convergent, 130 characters for carapace or dorsal spines treated as convergent, 129 characters for strengthening rays treated as convergent). Character descriptions and scorings are available on MorphoBank[67] (www.morphobank.org, https://doi.org/10.7934/P4146).

Maximum Parsimony analyses were run using TNT v1.5[68] using implied weights (concavity constant $k = 3$) and New Technology. The shortest tree was required to be retrieved 100 times, using tree bisection−reconnection to swap a single branch at a time on the trees in the memory[69]. Bayesian Inference Phylogenetic analyses were undertaken in MrBayes (version 3.2.6)[70]. A Markov (Mk) model was implemented, each analysis ran four runs of four chains, with a 25% burnin. Convergence was assessed using Tracer v1.7.2 to compare

posterior distributions, standard deviations of split frequencies <0.01, effective sample size of all parameters > 200. Following ref. 36, we compared the 'maximise information' and 'minimise assumptions' strategies of ref. 71, with the latter varying parameters to potentially allow better fit of model to data (exact parameters for each strategy provided in nexus files, see *Data Availability* and *Code Availability*, and ref. 71). Each strategy ran for at least 20 million generations (maximum 25 million generations). The posterior sample of optimal trees was visualised in multidimensional treespace[36], and bipartitions including the Castle Bank specimens were investigated. For analyses considering the impact of convergence of some morphological characters on our results, only the 'maximise information' strategy was used. The multidimensional treespace was constructed from the posterior samples of all four analyses considering some level of convergent evolution and the posterior sample of the 'maximum information' analysis that treated all characters as homologous and used only one Castle Bank terminal.

## Nomenclatural acts

This published work and the nomenclatural acts it contains have been registered in ZooBank, the proposed online registration system for the International Code of Zoological Nomenclature (ICZN). The ZooBank LSIDs (Life Science Identifiers) can be resolved and the associated information viewed through any standard web browser by appending the LSID to the prefix "http://zoobank.org/". The LSIDs for this publication are: publication: 6216E87D-6FC9-4A32-B5CF-5E8EF3D13440; act (genus): 9860A52F-4B3F-4B6F-AE96-CC090DB51046; act (species): 9F00C780-C781-4EDB-9C9C-52BA92F92171.

## Reporting summary

Further information on research design is available in the Nature Research Reporting Summary linked to this article.

## Data availability

Data files are available at MorphoBank (www.morphobank.org, https://doi.org/10.7934/P4146) and in the Open Science Framework (https://osf.io/, https://doi.org/10.17605/OSF.IO/4FTZY). Specimens are accessioned at Amgueddfa Cymru—National Museum Wales, Cardiff, UK (NMW).

## Code availability

R code, nexus files and tnt files for treespace and phylogenetic analyses have been uploaded to the Open Science Framework (https://osf.io/, https://doi.org/10.17605/OSF.IO/4FTZY).

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

## Acknowledgements

The support of the landowners, Ben Douel and family, and contributors to a crowdfunding appeal (including a Holloway Bursary from the Warwickshire Geological Conservation Society) that allowed the purchase of

photomicroscope equipment used in this study, is gratefully acknowledged. Funding was provided by a University of Cambridge Herchel Smith Postdoctoral Fellowship (SP), Chinese Academy of Sciences PIFI fellowships (2020VCB0014 and 2018VCB0014, JPB and LAM respectively) and National Science Foundation DEB #1856679 (JMW). Harriet B. Drage offered advice on figure presentation, and Lucy M. E. McCobb provided curatorial assistance.

## Author contributions

Paper written by S.P. with contributions and critical insights from all authors. J.P.B. and L.A.M. constructed Fig. 1, S.P. all other figures. J.P.B. and L.A.M. conducted fieldwork. All authors interpreted specimens. S.P., J.P.B. and L.A.M. conducted photography. S.P. and J.M.W. updated and coded taxa in character matrix. S.P. led phylogenetic and treespace analyses with input from J.M.W.

## Competing interests

The authors declare no competing interests
