## [Peer Review File · Nature Communications]

Ordovician opabiniid-like animals and the role of the proboscis in euarthropod head evolutionReviewers' Comments:

Reviewer #1:

Remarks to the Author:

This manuscript reports an opabiniid-like euarthropod from a new locality, described as bearing a frontal head structure, the proboscis. This species is suggested to occupy a phylogenetic position crownwards of opabiniids, sister to radiodonts and deuteropods, and therefore may represent a transitional taxon to interpret the head evolution in the euarthropods. The phylogenetic analyses and discussion are reasonable. I appreciate the authors considered two possibilities concerning the new specimens, both represented two ontogenetic stages of one species, and two distinct taxa. They also consider some controversial deuteropod taxa, *Kylinsia* and *Parapetoia* would change the topology of taxa stemwards of radiodonts. They got broad agreement in the results, consistently support a topology of a paraphyletic grade of lower group euarthropods with fused protocerebral appendages.

Below, I raise a few questions and concerns that I believe should be addressed prior to publication, largely focused on the morphological interpretation of the new taxon. Once these points are addressed suitably, I believe that this discovery would make a valuable contribution to our knowledge of early stages of arthropod evolution.

Line 65: Rather than 'Panarthropoda', I suggest to use "Euarthropoda". The new species was assigned into lower stem group euarthropod in the phylogenetic result, and it contributed to understanding the head evolution in Euarthropoda. Also, use 'euarthropod' instead of 'panarthropod' in line 69.

Line 69: The two specimens collected from a totally new Burgess Shale-type (BST) fauna, Castle Bank fauna (Builth Inlier, Wales, UK) from the Middle Ordovician. We know nothing about this deposit. So, it would help a lot to include a brief introduction of the new fauna, to help us evaluate if this BST deposit is of high fossil fidelity to support the morphological interpretation presented here. Specifically, the diagnostic features of opabiniid described seem not very easy to be recognized, at least based on the images only (Figs. 1-5). It doesn't mean the morphological interpretation by the authors are not convincing, but including more information on the preservation (fossil fidelity) and the faunal composition of the new deposit seems necessary.

Line 88: I'm unconvinced by the presence of a dorsal sclerite in the holotype specimen. In Fig. 1a, it is difficult to see much except the rounded margin. In Fig. 4a, it is equally plausible that the structure labelled 'ls' and 're' are sclerite and its extension.

Line 93: The interpretation of the oral plates is more or less okay (Fig. 1a, also see Fig. S2), but the so-called j-shaped gut isn't particularly convincing (Fig. 1a).

Line 107: The authors interpreted the 'darker region' preserves lines as the setal blades. Normally, the darker carbonaceous films (BST fossil) represent the very soft tissues, such as the eyes, and the internal soft tissues, nervous system, gut and the digestive glands. Also, are the details of the trunk appendages illustrated in Fig. 3 over-interpreted?

Line 128: The authors described the smallest specimen (3mm in length) (see Fig. 4) as a larva of *Mieridduryn*. It is possible but not conclusive, especially the authors interpreted the specimen as a molt rather than a carcass (Line 153). What's to say it is not the organic pieces preserved with some artefact structures, which are common in the collection of BST deposits.

Line 178: That is so interesting to know that *Parapeytoia* has been suggested to occupy a pivotal position between radiodonts and deuteropods, undermines support for *Kylinxia* as sister group to all other deuteropods. It is also great to see the phylogenetic analyses considering *Parapeytoia* frontal appendages as protocerebral and deutocerebral respectively indicate that its phylogenetic position is not driven by the segmental affinity of the frontal appendages (Figs. S8 and S9). I am wondering if a

more conservative interpretation of Mieridduryn, requiring certain characters to be coded differently (or simply as question marks) will potentially result in a different tree topology.

Line 260/278: The authors concluded that the proboscis (considered protocerebral in origin) may have been reduced to become the labrum (treated as deutocerebral) of deuteropods. It is hard for me to interpret a segmental transformation of the appendages from protocerebral to deutocerebral between opabiniids and deuteropods.

Reviewer #2:

Remarks to the Author:

This is a fascinating and important treatment of two new euarthropod specimens from the Ordovician of Wales which the authors interpret as opabiniid-like creatures with a single, flexible pre-oral proboscis. This would represent a major discovery in and of itself, extending the unique opabiniid morphology some 60 million years from the Cambrian into the Ordovician. However, the authors conduct a phylogenetic analysis that indicates the possession of a proboscis may have been plesiomorphic for deuteropods (Crown group arthropods and their immediate stem lineage) and that a reduction of the proboscis may have led to the formation of the arthropod labrum. This is an intriguing and, to my mind, important hypothesis.

In cases such as this everything hangs on the interpretation of the specimens, and here the specimens are incredibly difficult to interpret mainly due to their incredibly small size. The authors do a very good job of photographing the material and presenting their interpretations, and I do believe that their interpretations are convincing. While not complete, the possession of the proboscis, oral cone, and tail fan (in one of the specimens) does look verifiable. The phylogenetic treatment is also comprehensive and robust; in summary, I feel the interpretations of the specimens and their potential ramifications for arthropod evolution are justified.

I have just a few minor points or suggestions that the authors may wish to consider:

1. Regarding the second specimen, the main distinction between it and the holotype of Mieridduryn appears to be the multi-partite head sclerite, which the authors note may be an ontogenetic feature. However, the authors also indicate that the smaller specimen is likely to be a molt. I feel the authors should at least discuss the possibility that the sclerite could be broken or folded because of activity during the molting process, or potentially due to the compaction of a hollow molt by sediment during the preservation process.

2. Within the phylogeny the authors have included another opabiniid, *Utaurora*. Unless I am mistaken that genus name has not yet been published, and the authors may want to insert a different placeholder name for the taxon.

3. I think the authors may want to be a bit clearer about their hypothesis that the protocerebral appendages are secondarily unfused in radiodonts. I spent a little time while writing this review thinking it may be equally parsimonious for the appendages to have fused independently in opabiniids and the Castle Bank taxa; this is not the case (as shown clearly in Figure 6). Might it be worthwhile doing a simple ancestral state reconstruction to demonstrate this? I think the problem is that this is only I think explicitly stated in the conclusion, and it may want to be included in the discussion also.

4. The treatment of the position of megacheirans is somewhat contradictory, particularly in the introduction. In line 47 the authors cite a single paper suggesting that the innervation of the frontal appendages in megacheirans may be protocerebral rather than deutocerebral; however, there are a number of papers that strongly indicate the frontal appendages are deutocerebral. There is also regular mention of megacheirans being upper stem-group euarthropods (as in line 49). However, the

position of megacheirans fluctuates in analyses between the euarthropod stem and crown, with a number of analyses placing them within the chelicerate stem group. This position is also borne out by a number of the authors' own analyses (although indeed in some topologies the position of megacheirans is uncertain). The introduction and some of the discussion of megacheirans may want to be reworded to indicate the crown-group possibility. I realise however that this issue is incredibly complex and often changing, so discussing the issue without going off on a total tangent is hard. Potentially a single mention of the alternative possibilities for Megacheira would suffice.

Reviewer #3:

Remarks to the Author:

The authors describe two fossil specimens with a putative affinity to Opabinia. The fossils are flat and not well preserved. Nevertheless, the authors use these for far reaching conclusions about the evolution of the arthropod labrum. They claim that the proboscis is a fused frontal appendage connected to the protocerebrum. Evolutionarily, it gives rise to the labrum, which is also considered as being a fused appendage situated at the anterior end of the head. Their cladistics analysis resolves opabiniids as being paraphyletic with the new finds being more closely related to radiodonts and crown euarthropods than to Opabinia.

The manuscript faces several problems. One is, too many assumptions are taken for granted. The other is the poor condition of the fossils.

What is the evidence that the proboscis is a fused paired protocerebral appendage? Even in Opabinia the fusion is mere speculation just based on the bilaterally symmetric anterior parts. Furthermore, the pre-ocular position has not been proven. A look at a grasshopper's head shows antennae anterior to the eyes (as in Opabinia) but these antennae originate from the post-ocular deutocerebrum.

The view that the labrum originated from the frontal appendage is central for the evolutionary scenario put forward by the authors. However, there are several alternative views about the origin of the labrum. These are not even mentioned not to speak of a proper discussion. Likewise, the relative position and the putative homologues of the frontal appendage within panarthropods and euarthropods has been interpreted differently. Again, these various views are not mentioned, although they have an impact on the plausibility of the conclusions of the authors.

According to the cladistic analysis of the authors, the paired frontal appendages of Radiodonta would be a case of reverse evolution. Since this is a quite unexpected result, it would need a more careful treatment. The authors just mention that it is an apomorphy of radiodonts. However, why should Radiodonta reach secondarily the same character state as Pambdelurion. The same is true for the position of the mouth. Radiodonta gave up the posterior facing mouth. Hence, the implausibility of the parallel reversal of two independent characters cries for an explanation.

The newly described fossils are in a relatively poor condition and do not contain much information. The proboscis is too short to reach the putative mouth. The authors claim that the terminal ends are missing, but what is the evidence for this assumption? Since it is a lateral view, the proboscis could be a paired appendage. The position of the mouth opening is doubtful, to say the least. In general, most of the morphological characteristics are extrapolations of the situation in Opabinia.

REVIEWER COMMENTS

Reviewer #1 (Remarks to the Author):

This manuscript reports an opabiniid-like euarthropod from a new locality, described as bearing a frontal head structure, the proboscis. This species is suggested to occupy a phylogenetic position crownwards of opabiniids, sister to radiodonts and deuteropods, and therefore may represent a transitional taxon to interpret the head evolution in the euarthropods. The phylogenetic analyses and discussion are reasonable. I appreciate the authors considered two possibilities concerning the new specimens, both represented two ontogenetic stages of one species, and two distinct taxa. They also consider some controversial deuteropod taxa, *Kylinxia* and *Parapeytoia* would change the topology of taxa stemwards of radiodonts. They got broad agreement in the results, consistently support a topology of a paraphyletic grade of lower group euarthropods with fused protocerebral appendages.

Below, I raise a few questions and concerns that I believe should be addressed prior to publication, largely focused on the morphological interpretation of the new taxon. Once these points are addressed suitably, I believe that this discovery would make a valuable contribution to our knowledge of early stages of arthropod evolution.

We thank the reviewer for their constructive and broadly positive comments on our manuscript. We have provided additional photographs to support our morphological interpretation, including some using fluorescence, and conducted additional analyses to test the robustness of our results.

Line 65: Rather than 'Panarthropoda', I suggest to use "Euarthropoda". The new species was assigned into lower stem group euarthropod in the phylogenetic result, and it contributed to understanding the head evolution in Euarthropoda. Also, use 'euarthropod' instead of 'panarthropod' in line 69.

The systematic palaeontology should only include monophyletic groups. As Mieridduryn falls within the lower stem group of Euarthropoda, we cannot use 'Phylum Euarthropoda' as it is not in the crown. Neither can we denote 'stem group Euarthropoda' as this is paraphyletic, not monophyletic (One author, SP, has incorrectly used 'stem group Euarthropoda' in the past).

Line 69: The two specimens collected from a totally new Burgess Shale-type (BST) fauna, Castle Bank fauna (Builth Inlier, Wales, UK) from the Middle Ordovician. We know nothing about this deposit. So, it would help a lot to include a brief introduction of the new fauna, to help us evaluate if this BST deposit is of high fossil fidelity to support the morphological interpretation presented here. Specifically, the diagnostic features of opabiniid described seem not very easy to be recognized, at least based on the images only (Figs. 1-5). It doesn't mean the morphological interpretation by the authors are not convincing, but including more information on the preservation (fossil fidelity) and the faunal composition of the new deposit seems necessary.

The preservation is exceptional, and includes soft tissues and appendages of numerous phyla, including arthropods. A 'discovery' style paper for this fauna is in progress.

Action: We have included a brief introduction to the fauna [Geological setting subheading, Lines 73-85], and a new figure (figure 1) that shows the locality and local geology. We also

have supplied a figure solely for the referees (not for publication) that shows another total group euarthropod with carapace and appendages preserved.

Line 88: I'm unconvinced by the presence of a dorsal sclerite in the holotype specimen. In Fig. 1a, it is difficult to see much except the rounded margin. In Fig. 4a, it is equally plausible that the structure labelled 'ls' and 're' are sclerite and its extension.

We have supplied additional photographs in a new supplementary figure to show the dorsal sclerite in more detail (along with the gut – see below). We have included the possibility that 're' is the extension of the lateral sclerite on the second specimen in the main text.

Action: new supplementary figure S2 showing dorsal sclerite of holotype in more detail. Addition of possibility that rectangular elements are broken parts of lateral sclerites in main text. [lines 181-182]

Line 93: The interpretation of the oral plates is more or less okay (Fig. 1a, also see Fig. S2), but the so-called j-shaped gut isn't particularly convincing (Fig. 1a).

We thank the reviewer for supporting the interpretation of the oral plates in the mouth. For the gut, one issue may have been with our terminology, as only the distal part of the gut is visible (so the curved part of the 'j' rather than the whole 'j').

Action: We have refined the terminology here, using 'twists ventrally where it connects to the mouth' instead of 'j-shaped' as only the distal part of the gut which recurves and leads to the mouth is present. We have also provided additional images of the gut, included as a figure in the supplementary information (same figure as shows the anterior sclerite, figure S2). [Lines 116-117]

Line 107: The authors interpreted the 'darker region' preserves lines as the setal blades. Normally, the darker carbonaceous films (BST fossil) represent the very soft tissues, such as the eyes, and the internal soft tissues, nervous system, gut and the digestive glands.

*Darker regions have been interpreted as particularly organic rich areas, including internal soft tissues and nervous tissues, however multiple lines of evidence go into these interpretations. Furthermore, some darker regions in BST fossils have been interpreted not as internal tissues – because alternative interpretations are favoured by the morphology. In this study, the hair-like nature of the dark strands protruding from behind the flap support the identification of these features as setal structures. Similar hair-like features have been interpreted as setal structures in opabiniids (e.g. dorsal hair-like features in *Utaurora*, see Pates et al. 2022). In radiodonts dark lobate structures with fine hairs have been similarly described (for example in *Anomalocaris* and *Cambroraster* –Daley & Edgecombe 2014 fig. 10.5 and Moysiuk & Caron 2019 sup figs. 1B, 2B – note these are referred to as bands of lamellae in the latter publication). Faint dark lines running parallel to the long axis of the flap have been reported in radiodonts, described as strengthening rays (e.g. *Buccaspinea*, see Pates et al. 2021). In the Castle Bank material, we interpret overlying setal structures and strengthening rays, visible in this first flap, hence why some lines run parallel to the long axis while others are finer and run parallel to the short axis of the flap.*

Action: We have added a short additional explanation of how we interpret these features to the Further Remarks section of the supplementary materials, with reference to the papers mentioned in the reply above.

Also, are the details of the trunk appendages illustrated in Fig. 3 over-interpreted?

While we observed all features in Fig. 1 and Fig. 3 under the microscope, we appreciate that not all are completely visible in the accompanying photograph. We have adjusted figure 3 to remove some of the details of the trunk appendages that are not visible in the photograph.

Action: altered both figures 1 and 3 (now figures 2 and 4) to remove lines and labels for some details that are not clear in the photographs e.g. some of the annulations.

Line 128: The authors described the smallest specimen (3mm in length) (see Fig. 4) as a larva of *Mieridduryn*. It is possible but not conclusive, especially the authors interpreted the specimen as a molt rather than a carcass (Line 153). What's to say it is not the organic pieces preserved with some artefact structures, which are common in the collection of BST deposits.

We consider two possibilities for the smallest specimen – either an earlier ontogenetic stage of the holotype of Mieridduryn, or a second distinct taxon. We offer the possibility that the specimen is a moult (it is wrinkled and appears torn towards the anterior of the trunk), but do not rule out the possibility that it is a carcass.

Action: We have added the words 'though this is not conclusive' to the sentence where we suggest this specimen may represent a moult rather than a carcass [lines 177-8].

Action 2: We have added the word 'Alternatively' to the start of the paragraph where we discuss that this specimen may be an earlier stage of Mieridduryn than the holotype [line 189] This makes it clearer that we are discussing two possibilities rather than prioritising one over the other. At this stage we do not have enough data to be conclusive on this point.

Line 178: That is so interesting to know that *Parapeytoia* has been suggested to occupy a pivotal position between radiodonts and deuteropods, undermines support for *Kylinxia* as sister group to all other deuteropods. It is also great to see the phylogenetic analyses considering *Parapeytoia* frontal appendages as protocerebral and deutocerebral respectively indicate that its phylogenetic position is not driven by the segmental affinity of the frontal appendages (Figs. S8 and S9). I am wondering if a more conservative interpretation of *Mieridduryn*, requiring certain characters to be coded differently (or simply as question marks) will potentially result in a different tree topology.

It was very rewarding to hear that our strategy testing multiple hypotheses relating to the segmental affinity of the frontalmost appendages was of interest and value to the reviewer. In relation to the last point – many characters were coded conservatively (full details of coding are available in Morphobank). Note that some of the characters which the reviewer was less convinced by ('re' and gut) are not included as characters in the analysis.

We found the reviewers final comment particularly interesting, and have explored scenarios where characters are coded as convergently evolved between radiodonts and the Castle Bank specimens, to see what needs to change in the coding to result in a different tree topology.

We focused on the following three characters:

- *Sclerites in the head region*

- *Strengthening rays in the flaps*
- *Dorsal spines on the proboscis*

Our results show that treating some or all of these characters as convergent (i.e. not homologous) between radiodonts and Castle Bank specimens leads to a different tree topology that recovers a monophyletic Opabiniidae composed of Opabinia, Mieridduryn and Utaurora. We have added additional figures and new section to the discussion of these results, as a convergent origin of these features cannot be discounted and it has implications for our understanding of the evolution of euarthropods. Our aim is to be as thorough as possible in interrogating our data and underlying assumptions.

Action: Four additional phylogenetic analyses were run, and the posterior samples visualised in treespace. Analyses treating some or all of these characters as having evolved independently within the Castle Bank specimens recover a monophyletic Opabiniidae that includes the Castle Bank specimens – not a paraphyletic grade of proboscis-bearing euarthropods as in the main analyses treating these characters as homologous. We have included a new section in the discussion where we place these new results into context, and present full results as supplementary figures. [Discussion subsection: ‘Convergent characters, monophyletic opabiniids’, lines 343-374, figures S11, S12].

Line 260/278: The authors concluded that the proboscis (considered protocerebral in origin) may have been reduced to become the labrum (treated as deutocerebral) of deuteropods. It is hard for me to interpret a segmental transformation of the appendages from protocerebral to deutocerebral between opabiniids and deuteropods.

The reviewer is correct, that we suggest that the protocerebral proboscis may have reduced to become the labrum. The reviewer is mistaken, however, in suggesting that we treat the labrum as deutocerebral. We treat the labrum as protocerebral in origin for early deuteropods (see Lines 42-47 in introduction, and character matrix in Morphobank). This protocerebral origin is supported by both palaeoneurological data and developmental studies (reviewed for example by Ortega-Hernández et al. 2017; Jocksuch 2017). Thus, our proposed scenario does not require segmental transformation of the appendages – indeed there is no evidence for a deutocerebral segment in the lower stem group of Euarthropoda. The more complex heads of upper stem group and crown group euarthropods may relate to a subdivision of a single head segment into three ‘post gnathal segments’, rather than integration of trunk segments into the head (Lev et al. 2022).

Reviewer #2 (Remarks to the Author):

This is a fascinating and important treatment of two new euarthropod specimens from the Ordovician of Wales which the authors interpret as opabiniid-like creatures with a single, flexible pre-oral proboscis. This would represent a major discovery in and of itself, extending the unique opabiniid morphology some 60 million years from the Cambrian into the Ordovician. However, the authors conduct a phylogenetic analysis that indicates the possession of a proboscis may have been plesiomorphic for deuteropods (Crown group arthropods and their immediate stem lineage) and that a reduction of the proboscis may have led to the formation of the arthropod labrum. This is an intriguing and, to my mind, important hypothesis.

In cases such as this everything hangs on the interpretation of the specimens, and here the specimens are incredibly difficult to interpret mainly due to their incredibly small size. The authors do a very good job of photographing the material and presenting their interpretations, and I do believe that their interpretations are convincing. While not complete, the possession of the proboscis, oral cone, and tail fan (in one of the specimens) does look verifiable. The phylogenetic treatment is also comprehensive and robust; in summary, I feel the interpretations of the specimens and their potential ramifications for arthropod evolution are justified.

We thank the reviewer for their positive review and constructive suggestions. We appreciate the positive feedback that the interpretations are convincing and that the phylogenetic treatment is robust and comprehensive.

I have just a few minor points or suggestions that the authors may wish to consider:

1. Regarding the second specimen, the main distinction between it and the holotype of *Mieridduryn* appears to be the multi-partite head sclerite, which the authors note may be an ontogenetic feature. However, the authors also indicate that the smaller specimen is likely to be a molt. I feel the authors should at least discuss the possibility that the sclerite could be broken or folded because of activity during the molting process, or potentially due to the compaction of a hollow molt by sediment during the preservation process.

This is an interesting interpretation which we had not considered initially. We thank the reviewer for raising it.

Action: We have added a statement highlighting this possibility to the Remarks section. [Lines 180-182]

2. Within the phylogeny the authors have included another opabiniid, *Utaurora*. Unless I am mistaken that genus name has not yet been published, and the authors may want to insert a different placeholder name for the taxon.

*We thank the reviewer for highlighting this. Fortunately the paper describing *Utaurora* is now published (<https://doi.org/10.1098/rspb.2021.2093>), so a placeholder name is no longer necessary.*

3. I think the authors may want to be a bit clearer about their hypothesis that the protocerebral appendages are secondarily unfused in radiodonts. I spent a little time while writing this review thinking it may be equally parsimonious for the appendages to have fused

independently in opabiniids and the Castle Bank taxa; this is not the case (as shown clearly in Figure 6). Might it be worthwhile doing a simple ancestral state reconstruction to demonstrate this? I think the problem is that this is only I think explicitly stated in the conclusion, and it may want to be included in the discussion also.

We thank the reviewer for highlighting this and have made this point explicit in the Discussion.

Action: added sentences to the discussion: 'These results also suggest that the protocerebral appendage secondarily divided in radiodonts. For these topologies, a single fusion of the protocerebral appendage in opabiniids and subsequent un-fusion in radiodonts is slightly more parsimonious than the alternative: multiple independent fusions of the protocerebral appendage in opabiniids, the Castle Bank specimens, and deuteropods (Figure 9)' [Lines 322-326].

4. The treatment of the position of megacheirans is somewhat contradictory, particularly in the introduction. In line 47 the authors cite a single paper suggesting that the innervation of the frontal appendages in megacheirans may be protocerebral rather than deutocerebral; however, there are a number of papers that strongly indicate the frontal appendages are deutocerebral. There is also regular mention of megacheirans being upper stem-group euarthropods (as in line 49). However, the position of megacheirans fluctuates in analyses between the euarthropod stem and crown, with a number of analyses placing them within the chelicerate stem group. This position is also born out by a number of the authors' own analyses (although indeed in some topologies the position of megacheirans is uncertain). The introduction and some of the discussion of megacheirans may want to be reworded to indicate the crown-group possibility. I realise however that this issue is incredibly complex and often changing, so discussing the issue without going off on a total tangent is hard. Potentially a single mention of the alternative possibilities for Megacheira would suffice.

We appreciate this comment and the reviewer's constructive suggestion to provide mention for the alternative possibilities for Megacheira. As the affinities of Megacheira are beyond the scope of our study and analyses, we chose to alter the wording here to reflect the two possibilities – that megacheirans have been suggested to be upper stem group or crown group euarthropods.

Action: We have changed the phrase 'upper stem group euarthropods' to 'fossil deuteropods (upper stem group + crown group euarthropods)' at this point in the introduction [Lines 51-2].

Reviewer #3 (Remarks to the Author):

The authors describe two fossil specimens with a putative affinity to *Opabinia*. The fossils are flat and not well preserved. Nevertheless, the authors use these for far reaching conclusions about the evolution of the arthropod labrum. They claim that the proboscis is a fused frontal appendage connected to the protocerebrum. Evolutionarily, It gives rise to the labrum, which is also considered as being a fused appendage situated at the anterior end of the head. Their cladistics analysis resolves opabiniids as being paraphyletic with the new finds being more closely related to radiodonts and crown euarthropods than to *Opabina*. The manuscript faces several problems. One is, too many assumptions are taken for granted. The other is the poor condition of the fossils.

We thank the reviewer for taking the time to provide comments on our manuscript.

What is the evidence that the proboscis is a fused paired protocerebral appendage? Even in *Opabinia* the fusion is mere speculation just based on the bilaterally symmetric anterior parts. Furthermore, the pre-ocular position has not been proven. A look at a grasshopper's head shows antennae anterior to the eyes (as in *Opabinia*) but these antennae originate from the post-ocular deutocerebrum.

The hypothesis that the proboscis of Opabinia represents a fused pair of appendages has long been established in the literature. The earliest record we could find was Bergström (1986), but see also the character comparison of Hou et al. (1995, fig. 20 caption 4c). This inference has found further support from studies in the last 10 years, for example:

- 1) Palaeoneurological data of lower stem group relatives: No neurological tissues have been reported from an opabiniid, however neural tissues are known from the gilled lobopodian *Kerygmachela* (Park et al. 2018) which falls almost immediately stemwards of opabiniids in our results, and the radiodont *Lyrarapax* (Cong et al. 2014) crownwards of opabiniids. Thus, given the position of opabiniids in the euarthropod lower stem, a protocerebral origin for the proboscis is the most likely explanation.*
- 2) Short head that lacks any other differentiated appendages: Members of the euarthropod lower stem group have only one pair of specialised head appendages, innervated by the protocerebrum. Indeed, the brain of *Kerygmachela* and *Lyrarapax* are unipartite – they have only a protocerebrum. Therefore the suggestion that the appendage of *Opabinia* is not protocerebral is very difficult to support. The origination of a second pair of head appendages innervated by the deutocerebrum is one of multiple synapomorphies which define Deuteropoda – the group comprising the upper stem and crown groups of Euarthropoda (e.g. Ortega-Hernández 2016). The grasshopper case raised by the reviewer is not a suitable comparison to our material, as it is a derived crown group member. A grasshopper has a highly differentiated head, including numerous specialised appendages, among them a labrum. *Opabinia* and *Mieridduryn* do not have any evidence of a labrum (in *Opabinia*, no evidence from ~40 specimens), nor any differentiated head appendage with the notable exception of the proboscis. For a detailed review of the evolution of the euarthropod head, see for example Ortega-Hernández et al. (2017, table 1 and figure 7) or for a slightly different view Lev et al. 2022.*
- 3) Developmental inference: Chipman (2015) placed the evolution of the euarthropod stem lineage into a developmental framework, reconstructing the germband of notable taxa (including *Opabinia*). Chipman concluded that 'the only reasonable*

*interpretation of this appendage [the proboscis] is as a fusion of the paired protocerebral appendages found in *K. kierkegardii* [Kerygmachela] and the more crownward anomalocaridids [radiodonts]'. While this paper is indeed speculative, it aligns with the paleoneurological data (point 1), external morphology (point 2), and phylogenetic relationships (our data).*

The view that the labrum originated from the frontal appendage is central for the evolutionary scenario put forward by the authors. However, there are several alternative views about the origin of the labrum. These are not even mentioned not to speak of a proper discussion. Likewise, the relative position and the putative homologues of the frontal appendage within panarthropods and euarthropods has been interpreted differently. Again, these various views are not mentioned, although they have an impact on the plausibility of the conclusions of the authors.

We devoted a number of sentences within the introduction and supply references for alternative hypotheses relating to the evolution of the arthropod head, and how they disagree with the majority of the literature (see Lines 47-58). Thus it is difficult to agree with the reviewer that these views are not mentioned.

These alternative views are not widely held – the concept that the labrum evolved from an anterior pair of appendages is by far the most common in the recent literature. See this quote from a recent review specifically centered on the evolution of the labrum: Budd (2021): 'A widely (although not universally) accepted model of arthropod head evolution postulates that the labrum ... evolved from an anterior pair of appendages homologous to the frontal appendages of onychophorans'. We are clear throughout the manuscript that we base our interpretation of the proboscises of the Castle Bank specimens as a fused pair of protocerebral appendages on a combination of developmental, palaeoneurological, and morphological data. Should other workers wish to reinterpret these data in light of new evidence, we have presented 11 figures illustrating our material that facilitates this.

We also considered multiple interpretations for the segmental affinity of Parapeytoia appendages in our phylogenetic analyses, to check whether this impacts on the results within the lower stem group. As demonstrated by our analyses, treating Parapeytoia appendages as proto- or deutocerebral does not impact on the paraphyletic grade of proboscis bearing euarthropods recovered.

Action: We have added additional references to these sentences in the introduction highlighting works that put forward alternative interpretations for the evolution of the euarthropod head, and highlighted some references that offer an alternative view for the fate of the frontal appendages. Additional references: Haug et al. 2012; Scholtz 2016; Aria 2022. [Lines 47-58]

According to the cladistic analysis of the authors, the paired frontal appendages of Radiodonta would be a case of reverse evolution. Since this is a quite unexpected result, it would need a more careful treatment. The authors just mention that it is an apomorphy of radiodonts. However, why should Radiodonta reach secondarily the same character state as *Pambdelurion*. The same is true for the position of the mouth. Radiodonta gave up the posterior facing mouth. Hence, the implausibility of the parallel reversal of two independent characters cries for an explanation.

The reviewer makes a number of errors in this statement. Firstly, we do not propose that the frontal appendages of Radiodonta are the same as those in Pambdelurion. The frontal

appendages of radiodonts are arthropodized, composed of articulating podomeres, and bear dorsal spines. Those of Pambdelurion are annulated and lack dorsal spines. They are similar in that they are paired and protocerebral. A reversal in one character is not unexpected, as character reversals and homoplasies are common throughout the history of life. Considering the other characters mentioned (arthropodization, dorsal spines) we suggest how the Castle Bank specimens may help to explain the evolution from an annulated appendage to one which is composed of podomeres with dorsal spines.

Secondly, the reviewer asks about the ventral mouth in radiodonts, compared to the posterior one in opabiniids. Ours is not the first study to recover opabiniids stemwards of radiodonts in the euarthropod stem lineage – this result has been recovered numerous times in phylogenetic analyses, and for this character matrix under Bayesian Inference methods (e.g. Pates et al. 2022; Howard et al. 2020 – the latter not by us). The alternative – that there was no reversal in the mouth position – requires opabiniids to be recovered crownwards of radiodonts, which is not supported by our results or by numerous previous studies using similar matrices with Bayesian Inference methods (see references above). This alternative is a result we previously retrieved before including the Castle Bank taxa (Pates et al. 2022), however, almost exclusively with maximum parsimony. When Castle Bank taxa are added, this result disappears completely, indicating that the new fossils provide support to resolve this very topology.

The reviewer states that our results are ‘implausible’. This reflects a misunderstanding of how our analyses were run. We did not set out with an aim to create a particular topology. Our results reflect our observations of the fossil material, primary homology hypotheses (i.e. character coding), and methodology use to test these hypotheses, which we then interpreted. We have subjected our morphological matrix to a series of phylogenetic analyses and tested multiple different models of evolution. All retrieved broadly similar results.

Nonetheless, we have run additional analyses as suggested by reviewer 1 to test what impact there was on the analyses if we considered some features of the Castle Bank specimens as convergent with those of radiodonts in the primary homology hypothesis (i.e. character coding). Considering some of the features common to the Castle Bank specimens (dorsal spines, strengthening rays, carapace) as convergent leads to the Castle Bank specimens as being recovered as opabiniids. We have included discussion of this further interrogation of our morphological matrix and its implications in the text.

Action: We ran a further four sets of phylogenetic analyses, which treated a number of characters as convergent between opabiniids and radiodonts. We then subjected the posterior samples to a multidimensional treespace analysis (the same as our initial matrices that assumed homology between opabiniids and radiodonts), visualising bipartitions including the Castle Bank specimens. These new analyses recovered the Castle Bank specimens within monophyletic Opabiniidae rather than in a paraphyletic grade as our primary analyses from the initial submission. We added a section to the discussion where we explore the implications of these results [lines 343-374 and figures S11, S12].

The newly described fossils are in a relatively poor condition and do not contain much information.

As outlined in the description, the fossils contain abundant information. The presence of annulations in the proboscis, setal structures, and plates in the oral cone, as well as blades in the caudal fan, all support a lower stem group affinity for these specimens. Our

interpretations are supported by the broadly positive comments of reviewers 1 and 2 (above).

The proboscis is too short to reach the putative mouth. The authors claim that the terminal ends are missing, but what is the evidence for this assumption?

The reviewer argues that the proboscis is both too short to reach the mouth but that it is also complete, arguing two contradictory points simultaneously. We only suggested that the proboscis in the smaller specimen is incomplete (and have added the word “likely” to further indicate this is a suggestion). Support for this comes from the change in fidelity of preservation along the length of the proboscis. The part closest to the head displays well preserved annulations, an internal canal, and dorsal spines. The more distal part only displays an outline, which grades into the surrounding matrix. Therefore the suggestion that the distal part is incomplete is supported by the preservation (see Figures 7, 8 in main text, and Figure S4 for multiple illumination conditions).

Action: Added word ‘likely’ to description of the proboscis to clarify that it is most likely incomplete [line 159]. Added additional photographs of the proboscis including using a new technique (for these fossils) fluorescence (Figure 8).

Since it is a lateral view, the proboscis could be a paired appendage.

While the smaller specimen is preserved in broadly lateral aspect, the oblique preservation of the larger specimen (holotype of Mieridduryn) clearly shows only one appendage in the head region. Furthermore, the reviewer’s claim that lateral preservation would obscure a second appendage is not well founded, as numerous radiodont fossils preserved in lateral aspect show multiple appendages. See for example Anomalocaris canadensis specimens in Daley & Edgecombe (2014, figs. 5, 6, 10A). The chances of superimposing both appendages even with perfect lateral compression are low, given that the appendages move independently and in radiodonts are sometimes partially disarticulated from the head.

The position of the mouth opening is doubtful, to say the least.

The position of the mouth opening is indicated by not only the oral cirlet but also its position at the terminus of the gut (Figure S2).

Action: We have included a new figure to further illustrate the position of the oral cirlet in relation to the gut (Figure S2)

In general, most of the morphological characteristics are extrapolations of the situation in *Opabinia*.

The reviewer is mistaken on this point. The misunderstanding of the reviewer is highlighted by the fact that most of our phylogenetic analyses do not even find a sister relationship between the Castle Bank fossils and Opabinia and/or Utaurora. If the morphological characteristics were purely extrapolated from Opabinia, we would expect these taxa to form a clade. The morphology is based on what we have observed in the fossil material, and supported by our numerous diagrams and line drawings.

References cited in review

- Aria, C., 2022. The origin and early evolution of arthropods. *Biological Reviews*, Early view. Doi:10.1111/BRV.12864
- Bergström, J., 1986. *Opabinia Anomalocaris*, unique Cambrian 'arthropods'. *Lethaia*, 19(3), pp.241-246.
- Budd, G.E., 2021. The origin and evolution of the euarthropod labrum. *Arthropod structure & development*, 62, p.101048.
- Chipman, A.D., 2015. An embryological perspective on the early arthropod fossil record. *BMC Evolutionary Biology*, 15(1), pp.1-18.
- Cong, P., Ma, X., Hou, X., Edgecombe, G.D. and Strausfeld, N.J., 2014. Brain structure resolves the segmental affinity of anomalocaridid appendages. *Nature*, 513(7519), pp.538-542.
- Daley, A.C. and Edgecombe, G.D., 2014. Morphology of *Anomalocaris canadensis* from the Burgess Shale. *Journal of Paleontology*, 88(1), pp.68-91.
- Haug, J.T., Briggs, D.E. and Haug, C., 2012. Morphology and function in the Cambrian Burgess Shale megacheiran arthropod *Leanchoilia superlata* and the application of a descriptive matrix. *BMC Evolutionary Biology*, 12(1), pp.1-20.
- Hou, X.G., Bergström, J. and Ahlberg, P., 1995. *Anomalocaris* and other large animals in the Lower Cambrian Chengjiang fauna of southwest China. *GFF*, 117(3), pp.163-183.
- Howard, R.J., Hou, X., Edgecombe, G.D., Salge, T., Shi, X. and Ma, X., 2020. A tube-dwelling early Cambrian lobopodian. *Current Biology*, 30(8), pp.1529-1536.
- Jockusch, E.L., 2017. Developmental and evolutionary perspectives on the origin and diversification of arthropod appendages. *Integrative and comparative biology*, 57(3), pp.533-545.
- Lev, O., Edgecombe, G.D. and Chipman, A.D., 2022. Serial Homology and Segment Identity in the Arthropod Head. *Integrative Organismal Biology*, 4(1), p.obac015.
- Moysiuk, J. and Caron, J.B., 2019. A new hurdiid radiodont from the Burgess Shale evinces the exploitation of Cambrian infaunal food sources. *Proceedings of the Royal Society B*, 286(1908), p.20191079.
- Ortega-Hernández, J., 2016. Making sense of 'lower' and 'upper' stem-group Euarthropoda, with comments on the strict use of the name Arthropoda von Siebold, 1848. *Biological Reviews*, 91(1), pp.255-273.
- Ortega-Hernández, J., Janssen, R. and Budd, G.E., 2017. Origin and evolution of the panarthropod head—a palaeobiological and developmental perspective. *Arthropod structure & development*, 46(3), pp.354-379.
- Park, T.Y.S., Kihm, J.H., Woo, J., Park, C., Lee, W.Y., Smith, M.P., Harper, D.A., Young, F., Nielsen, A.T. and Vinther, J., 2018. Brain and eyes of *Kerygmachela* reveal protocerebral ancestry of the panarthropod head. *Nature communications*, 9(1), pp.1-7.
- Pates, S., Lerosey-Aubril, R., Daley, A.C., Kier, C., Bonino, E. and Ortega-Hernández, J., 2021. The diverse radiodont fauna from the Marjum Formation of Utah, USA (Cambrian: Drumian). *PeerJ*, 9, p.e10509.

Pates, S., Wolfe, J.M., Lerosey-Aubril, R., Daley, A.C. and Ortega-Hernández, J., 2022. New opabiniid diversifies the weirdest wonders of the euarthropod stem group. *Proceedings of the Royal Society B*, 289(1968), p.20212093.

Scholtz, G., 2016. Heads and brains in arthropods: 40 years after the 'endless dispute'. *In* *Structure and Evolution of Invertebrate Nervous Systems* (eds A. Schmidt-Rhaesa, S. Harzsch and G. Purschke). Oxford University Press, Oxford.

Reviewers' Comments:

Reviewer #1:

Remarks to the Author:

I appreciate the authors' efforts on the manuscript revisions. The phylogenetic analyses are robust. They have also supplied one more bivalved arthropod and more figures to help us evaluate if the new deposit is of high fossil fidelity. I agree with the authors that it is an exceptional preserved deposit, and is of high potential to preserve the soft tissues. While the fossil preservation is not as good as we expected, the morphological interpretations are more or less convincing. I admit it is a bit difficult for me to interpret the specimens exposed from this new deposit, because in other exceptional preserved localities, such as Burgess Shale and Chengjiang, the anatomic features are much more distinguishable. Hopefully, more evidences from various taxa in the following research could strengthen the morphological interpretation presented here. Except for a bit concerns on the relatively poor condition of the fossils, I do not have any more questions on this manuscript.

Reviewer #2:

Remarks to the Author:

I appreciate the authors addressing my previous comments on the manuscript. I think the revised manuscript is a big improvement, and think the discussions on the possible homologous or convergent origins of the relevant structures in opabiniids and anomalocaridids/euarthropods will be of broad interest.

Reviewer #3:

Remarks to the Author:

The revised version shows some improvements. Alternative hypotheses about head segmentation are cited and alternative scenarios of morphological evolution are discussed. Nevertheless, the central problem of the manuscript is the gap between the quality of the fossil material and the far-reaching deductions about arthropod evolution. Neither the existence of dorsal spines nor an annulation of the proboscis are convincingly shown, not to speak of the various head sclerites and the posterior-facing mouth (which does not make functional sense, if there is no structure that stops the food flow anterior to the mouth opening). For instance, the structures labelled as spines in Figs. 3 and suppl. 1 can be anything and similar structures occur on the proboscis' opposite sides as well. In particular, if the small size of the two specimens is considered. The spines in Fig. 8 are even less convincing. Given that these spines play a major role for the central hypothesis of the authors, this is not enough evidence. If the characters are weak, then even the most sophisticated cladistics analysis is also weak.

Line 35: the protocerebrum is not the anteriormost "segment" of the brain, but the "neuromere" of the anteriormost segment.

Ine 43: in the light of the controversial views on head segmentation, "demonstrate" should be replaced by "suggest"

In their reply to my review the authors write:

The hypothesis that the proboscis of Opabinia represents a fused pair of appendages has long been established in the literature. The earliest record we could find was Bergström (1986), but see also the character comparison of Hou et al. (1995, fig. 20 caption 4c). This inference has found further support from studies in the last 10 years, for example:

1) Palaeoneurological data of lower stem group relatives: No neurological tissues have been reported from an opabiniid, however neural tissues are known from the gilled lobopodian *Kerygmachela* (Park et al. 2018) which falls almost immediately stemwards of opabiniids in our results, and the radiodont

Lyrarapax (Cong et al.

2014) crownwards of opabiniids. Thus, given the position of opabiniids in the euarthropod lower stem, a protocerebral origin for the proboscis is the most likely explanation.

2) Short head that lacks any other differentiated appendages: Members of the euarthropod lower stem group have only one pair of specialised head appendages, innervated by the protocerebrum. Indeed, the brain of *Kerygmachela* and *Lyrarapax* are unipartite – they have only a protocerebrum. Therefore the suggestion that the appendage of *Opabinia* is not protocerebral is very difficult to support. The origination of a second pair of head appendages innervated by the deutocerebrum is one of multiple synapomorphies which define Deuteropoda – the group comprising the upper stem and crown groups of Euarthropoda (e.g. Ortega-Hernández 2016). The grasshopper case raised by the reviewer is not a suitable comparison to our material, as it is a derived crown group member. A grasshopper has a highly differentiated head, including numerous specialised appendages, among them a labrum. *Opabinia* and *Mieridduryn* do not have any evidence of a labrum (in *Opabinia*, no evidence

from ~40 specimens), nor any differentiated head appendage with the notable exception of the proboscis. For a detailed review of the evolution of the euarthropod head, see for example Ortega-Hernández et al. (2017, table 1 and figure 7) or for a slightly different view Lev et al. 2022.

3) Developmental inference: Chipman (2015) placed the evolution of the euarthropod stem lineage into a developmental framework, reconstructing the germband of notable taxa (including *Opabinia*). Chipman concluded that ‘the only reasonable interpretation of this appendage [the proboscis] is as a fusion of the paired protocerebral appendages found in *K. kierkegardii* [*Kerygmachela*] and the more crownward anomalocaridids [radiodonts]’. While this paper is indeed speculative, it aligns with the paleoneurological data (point 1), external morphology (point 2), and phylogenetic relationships (our data).

The rebuttal to my skepticism about the nature of the proboscis involves the citation of some authorities (Bergström, Hou et al.) but not an argument. As the authors state, there are no palaeoneurological data and the attempt of Chipman to reconstruct Cambrian embryos is based on circular reasoning. Hence, the only reason to identify the proboscis as fused pair of protocerebral appendages is the conclusion, what else should it be. The same is true for the view that the proboscis is innervated by the protocerebrum. Both ideas are just deductions that are not tested by independent, additional evidence. Indeed, it may be a reasonable assumption that the proboscis derived from a paired appendage. But what if the proboscis is not serially homologous to segmental limbs, like e.g. the horns of some beetles and cicadas, the paragnaths of crustaceans, or the lateral horns of cirriped nauplii, which were interpreted by Darwin as limbs? The grasshopper example has been chosen, because it demonstrates that a superficial topology of structures does not necessarily reflect the embryonic arrangement of the primordia of these structures. It did not imply that *Opabinia* had a labrum or any other mouthpart of an insect.

These alternative views are not widely held – the concept that the labrum evolved from an anterior pair of appendages is by far the most common in the recent literature. See this quote from a recent review specifically centered on the evolution of the labrum: Budd (2021): ‘A widely (although not universally) accepted model of arthropod head evolution postulates that the labrum ... evolved from an anterior pair of appendages homologous to the frontal appendages of onychophorans’. We are clear throughout the manuscript that we base our interpretation of the proboscises of the Castle Bank specimens as a fused pair of protocerebral appendages on a combination of developmental, palaeoneurological, and morphological data. Should other workers wish to reinterpret these data in light of new evidence, we have presented 11 figures illustrating our material that facilitates this.

We also considered multiple interpretations for the segmental affinity of *Parapeytoia* appendages in our phylogenetic analyses, to check whether this impacts on the results within the lower stem group. As demonstrated by our analyses, treating *Parapeytoia* appendages as proto- or deutocerebral does not impact on the paraphyletic grade of proboscis bearing euarthropods recovered.

Action: We have added additional references to these sentences in the introduction highlighting works that put forward alternative interpretations for the evolution of the euarthropod head, and highlighted some references that offer an alternative view for the fate of the frontal appendages. Additional references: Haug et al. 2012; Scholtz 2016; Aria 2022. [Lines 47-58]

This rebuttal is also problematic. Again, an authority (Budd) is cited, who, of course, likes his own ideas best. The view that Budd's views are the most common in the recent literature is due to several reviews written by Budd and coworkers during the last years. This is also the reason for the view that they are widely accepted – they are not (and this is not just the feeling of the reviewer).

The reviewer makes a number of errors in this statement. Firstly, we do not propose that the frontal appendages of Radiodonta are the same as those in Pambdelurion. The frontal appendages of radiodonts are arthropodized, composed of articulating podomeres, and bear dorsal spines. Those of Pambdelurion are annulated and lack dorsal spines. They are similar in that they are paired and protocerebral. A reversal in one character is not unexpected, as character reversals and homoplasies are common throughout the history of life. Considering the other characters mentioned (arthropodization, dorsal spines) we suggest how the Castle Bank specimens may help to explain the evolution from an annulated appendage to one which is composed of podomeres with dorsal spines.

The sameness meant relates to the paired nature of the appendages. Character reversals are not that common and always problematic. The reappearance of paired appendages after a fusion needs an explanation. This is still missing in the revised version. See lines 322 to 326. Furthermore, if the secondary separation of the appendage evolved in the radiodont lineage, together with the articulation of the frontal appendages, then this arthropodization would be convergent to that of the Deuteropoda. Alternatively, the only arthropodized appendage would have lost its jointed character in the lineage leading to deuteropods, but at the same time arthropodization was transferred to the remaining limbs. This conflict deserves a detailed discussion.

Lines 334 ff.: Does this imply a deutocerebral nature of the paired frontal appendages of radiodonts or is this meant to explain the advantages of have paired anterior appendages? I guess the authors favour the latter but it should be more clearly expressed.

REVIEWER COMMENTS

Reviewer #1 (Remarks to the Author):

I appreciate the authors' efforts on the manuscript revisions. The phylogenetic analyses are robust. They have also supplied one more bivalved arthropod and more figures to help us evaluate if the new deposit is of high fossil fidelity. I agree with the authors that it is an exceptional preserved deposit, and is of high potential to preserve the soft tissues. While the fossil preservation is not as good as we expected, the morphological interpretations are more or less convincing. I admit it is a bit difficult for me to interpret the specimens exposed from this new deposit, because in other exceptional preserved localities, such as Burgess Shale and Chengjiang, the anatomic features are much more distinguishable. Hopefully, more evidences from various taxa in the following research could strengthen the morphological interpretation presented here. Except for a bit concerns on the relatively poor condition of the fossils, I do not have any more questions on this manuscript.

We thank the reviewer for their positive response to our revisions. We wish to reiterate our thanks for their comments on our work which have substantially improved it.

Reviewer #2 (Remarks to the Author):

I appreciate the authors addressing my previous comments on the manuscript. I think the revised manuscript is a big improvement, and think the discussions on the possible homologous or convergent origins of the relevant structures in opabiniids and anomalocaridids/euarthropods will be of broad interest.

We thank the reviewer for their comments, and for their suggestions that have broadened our discussion and scope. We thank them for their positive recommendations and appreciate that they think our expanded discussion will be of broad interest.

Reviewer #3 (Remarks to the Author):

The revised version shows some improvements. Alternative hypotheses about head segmentation are cited and alternative scenarios of morphological evolution are discussed. Nevertheless, the central problem of the manuscript is the gap between the quality of the fossil material and the far-reaching deductions about arthropod evolution. Neither the existence of dorsal spines nor an annulation of the proboscis are convincingly shown, not to speak of the various head sclerites and the posterior-facing mouth (which does not make functional sense, if there is no structure that stops the food flow anterior to the mouth opening). For instance, the structures labelled as spines in Figs. 3 and suppl. 1 can be anything and similar structures occur on the proboscis' opposite sides as well. In particular, if the small size of the two specimens is considered. The spines in Fig. 8 are even less convincing. Given that these spines play a major role for the central hypothesis of the authors, this is not enough evidence. If the characters are weak, then even the most sophisticated cladistics analysis is also weak.

We thank the reviewer for offering a second round of comments on the manuscript. While we disagree on how to interpret the material, and on broader scale views on euarthropod evolution, we appreciate the value of a diverse view in directing us where to strengthen our

arguments, and where to highlight alternative interpretations and scenarios. We have increased the resolution and incorporated additional structural labels for our figures (as suggested by the Editor) to make some of these disputed features more visible.

Line 35: the protocerebrum is not the anteriormost “segment” of the brain, but the “neuromere” of the anteriormost segment.

Thank you for highlighting this.

Action: added ‘neuromere of the’ to line 35. Sentence now reads: ‘... paired appendages innervated by the protocerebrum, the neuromere of the anteriormost segment of the brain.’.

Line 43: in the light of the controversial views on head segmentation, “demonstrate” should be replaced by “suggest”

We accept that we can rephrase this sentence to reflect multiple views on head segmentation in arthropods.

Action: we have rephrased this paragraph: ‘Developmental, morphological and neurological data support the interpretation that protocerebral appendages transformed through the euarthropod stem lineage, from paired annulated appendages in gilled lobopodians such as Kerygmachela, to a fused proboscis in opabiniids, to the arthropodized and sclerotized appendages of radiodonts ^{3,14,20,24,25}, and subsequently the fused labrum as seen in nearly all extant euarthropods ^{7,14,26} (but see e.g. Refs ^{27–29} for alternative views on the homology radiodont frontal appendages).

In their reply to my review the authors write:

The hypothesis that the proboscis of Opabinia represents a fused pair of appendages has long been established in the literature. The earliest record we could find was Bergström (1986), but see also the character comparison of Hou et al. (1995, fig. 20 caption 4c). This inference has found further support from studies in the last 10 years, for example:

1) Palaeoneurological data of lower stem group relatives: No neurological tissues have been reported from an opabiniid, however neural tissues are known from the gilled lobopodian Kerygmachela (Park et al. 2018) which falls almost immediately stemwards of opabiniids in our results, and the radiodont Lyrarapax (Cong et al.2014) crownwards of opabiniids. Thus, given the position of opabiniids in the euarthropod lower stem, a protocerebral origin for the proboscis is the most likely explanation.

2) Short head that lacks any other differentiated appendages: Members of the euarthropod lower stem group have only one pair of specialised head appendages, innervated by the protocerebrum. Indeed, the brain of Kerygmachela and Lyrarapax are unipartite – they have only a protocerebrum. Therefore the suggestion that the appendage of Opabinia is not protocerebral is very difficult to support. The origination of a second pair of head appendages innervated by the deutocerebrum is one of multiple synapomorphies which define Deuteropoda – the group comprising the upper stem and crown groups of Euarthropoda (e.g. Ortega-Hernández 2016). The grasshopper case raised by the reviewer is not a suitable comparison to our material, as it is a derived crown group member. A grasshopper has a highly differentiated head, including numerous specialised appendages, among them a labrum. Opabinia and Mieridduryn do not have any evidence of a labrum (in Opabinia, no evidence from ~40 specimens), nor any differentiated head appendage with the notable exception of the proboscis. For a detailed review of the evolution of the

euarthropod head, see for example Ortega-Hernández et al. (2017, table 1 and figure 7) or for a slightly different view Lev et al. 2022.

3) Developmental inference: Chipman (2015) placed the evolution of the euarthropod stem lineage into a developmental framework, reconstructing the germband of notable taxa (including Opabinia). Chipman concluded that ‘the only reasonable interpretation of this appendage [the proboscis] is as a fusion of the paired protocerebral appendages found in K. kierkegardii [Kerygmachela] and the more crownward anomalocaridids [radiodonts]’. While this paper is indeed speculative, it aligns with the paleoneurological data (point 1), external morphology (point 2), and phylogenetic relationships (our data).

The rebuttal to my skepticism about the nature of the proboscis involves the citation of some authorities (Bergström, Hou et al.) but not an argument. As the authors state, there are no palaeoneurological data and the attempt of Chipman to reconstruct Cambrian embryos is based on circular reasoning. Hence, the only reason to identify the proboscis as fused pair of protocerebral appendages is the conclusion, what else should it be. The same is true for the view that the proboscis is innervated by the protocerebrum. Both ideas are just deductions that are not tested by independent, additional evidence. Indeed, it may be a reasonable assumption that the proboscis derived from a paired appendage. But what if the proboscis is not serially homologous to segmental limbs, like e.g. the horns of some beetles and cicadas, the paragnaths of crustaceans, or the lateral horns of cirriped nauplii, which were interpreted by Darwin as limbs? The grasshopper example has been chosen, because it demonstrates that a superficial topology of structures does not necessarily reflect the embryonic arrangement of the primordia of these structures. It did not imply that Opabinia had a labrum or any other mouthpart of an insect.

In our original response we outlined three complementary lines of evidence that support the interpretation of the proboscis as a fused appendage. These form a coherent argument. While no palaeoneurological data are known from opabiniids, the concept of phylogenetic bracketing is well established, and so based on the protocerebral innervation of appendages in Kerygmachela and radiodonts it is possible to infer that the proboscis was innervated by the protocerebrum.

The reviewer remains sceptical, arguing that it is better to instead consider the proboscis as a novel structure. In our view this is not an appealing argument as:

1) the proboscis is annulated (and in the Castle Bank specimens also bears dorsal spines) – these are features shared with other protocerebral appendages, in lobopodians, lower stem group euarthropods such as Kerygmachela (annulations) and spines (radiodonts). These similarities argue against it being a novel structure without an appendicular origin.

2) Annulations are a feature of “sticking out” structures that have only been interpreted as appendages (e.g. lobopodians, modern onychophorans), providing additional support to interpret the proboscis as such. Horns and paragnaths lack such features.

3) Opabiniids and the Castle Bank taxa lack a paired appendage comparable to that of gilled lobopodians or radiodonts. So if the proboscis is considered a novel structure, then it also requires the simultaneous loss of protocerebral appendages (which are shared by onychophorans, Kerygmachela, and Lyrarapax). It is more parsimonious to consider these as homologous structures.

Action: We have added text to the discussion (lines 285-303) which sets out our reasoning for considering the proboscis as a fused pair of protocerebral appendages rather than as a

novel structure. We also state that if our interpretation is incorrect then 'it [the proboscis] would have limited implications for the evolution of the head in the remainder of the group'.

Action 2: We have added text to subsection Convergent characters, monophyletic opabiniids which explains why we think that the interpretation of the proboscis as homologous to frontal appendages will make less difference to the outcome for this scenario (lines 389-393).

These alternative views are not widely held – the concept that the labrum evolved from an anterior pair of appendages is by far the most common in the recent literature. See this quote from a recent review specifically centered on the evolution of the labrum: Budd (2021): 'A widely (although not universally) accepted model of arthropod head evolution postulates that the labrum ... evolved from an anterior pair of appendages homologous to the frontal appendages of onychophorans'. We are clear throughout the manuscript that we base our interpretation of the proboscises of the Castle Bank specimens as a fused pair of protocerebral appendages on a combination of developmental, palaeoneurological, and morphological data. Should other workers wish to reinterpret these data in light of new evidence, we have presented 11 figures illustrating our material that facilitates this.

We also considered multiple interpretations for the segmental affinity of Parapeytoia appendages in our phylogenetic analyses, to check whether this impacts on the results within the lower stem group. As demonstrated by our analyses, treating Parapeytoia appendages as proto- or deutocerebral does not impact on the paraphyletic grade of proboscis bearing euarthropods recovered.

Action: We have added additional references to these sentences in the introduction highlighting works that put forward alternative interpretations for the evolution of the euarthropod head, and highlighted some references that offer an alternative view for the fate of the frontal appendages. Additional references: Haug et al. 2012; Scholtz 2016; Aria 2022. [Lines 47-58]

This rebuttal is also problematic. Again, an authority (Budd) is cited, who, of course, likes his own ideas best. The view that Budd's views are the most common in the recent literature is due to several reviews written by Budd and coworkers during the last years. This is also the reason for the view that they are widely accepted – they are not (and this is not just the feeling of the reviewer).

The reviewer argues that referring to the recent peer-reviewed literature is problematic. While others may hold the same views as this reviewer in private, we can only base our citations on what has been published; as yet, these views are the dominant perspective in the literature. For example, other workers who are not collaborators with Budd also support that the labrum evolved from a pair of frontal appendages homologous to the frontal appendages of onychophorans, while disagreeing with Budd (2021) on the segmental affinities of the appendages of some megacheirans. See for example Lan et al. (2021, figure 4). We also note that none of the authors on this manuscript have co-authored with Budd.

If future publications change this balance, and this impacts the wider interpretation of the labrum origins (including the evidence presented here), then that will be a natural progression of the science, but we cannot pre-empt that process before those interpretations are published.

Action: we have previously increased the number of references to this part of the introduction as part of the last round of revisions. The reviewer has not highlighted any further studies that we should include here, so we have made no additional changes.

The reviewer makes a number of errors in this statement. Firstly, we do not propose that the frontal appendages of Radiodonta are the same as those in Pambdelurion. The frontal appendages of radiodonts are arthropodized, composed of articulating podomeres, and bear dorsal spines. Those of Pambdelurion are annulated and lack dorsal spines. They are similar in that they are paired and protocerebral. A reversal in one character is not unexpected, as character reversals and homoplasies are common throughout the history of life. Considering the other characters mentioned (arthropodization, dorsal spines) we suggest how the Castle Bank specimens may help to explain the evolution from an annulated appendage to one which is composed of podomeres with dorsal spines.

The sameness meant relates to the paired nature of the appendages. Character reversals are not that common and always problematic. The reappearance of paired appendages after a fusion needs an explanation. This is still missing in the revised version. See lines 322 to 326. Furthermore, if the secondary separation of the appendage evolved in the radiodont lineage, together with the articulation of the frontal appendages, then this arthropodization would be convergent to that of the Deuteropoda. Alternatively, the only arthropodized appendage would have lost its jointed character in the lineage leading to deuteropods, but at the same time arthropodization was transferred to the remaining limbs. This conflict deserves a detailed discussion.

We appreciate the reviewer's request for additional discussion of this point. We have added discussion in two places to expand on our ideas and place them into a developmental and evolutionary context. Note that homeotic transformation of appendages has been established through genetic work, indeed appendages can be very significantly transformed with only small genetic changes – see Averof & Patel (1997).

Action: We have added the following to this paragraph: 'This topology also requires that arthropodized appendages arose convergently in radiodonts and deuteropods (Figure 11). The regularly spaced dorsal spines may have provided a repeated pattern along the proximodistal axis that might have been co-opted during development for joint formation⁵² in the frontal appendages of radiodonts (Supplementary Discussion). In deuteropods, this same repeated pattern may have been co-opted by the deutocerebral appendage following the expansion of the head into multiple segments and sub-functionalization of head appendages²³.' (lines 344-350)

Action 2: We have added the following to the end of the subsection Homologous characters, paraphyletic grade of proboscis-bearing lower stem group euarthropods:

'Thus after subdivision of the head, the appendages of each segment evolved independently specialising for distinct functions²³, with the genetic basis of both a fused labrum and repeated pattern for joints both potentially present in the proboscis of the Castle Bank specimens.' (lines 366-369)

Lines 334 ff.: Does this imply a deutocerebral nature of the paired frontal appendages of radiodonts or is this meant to explain the advantages of have paired anterior appendages? I guess the authors favour the latter but it should be more clearly expressed.

Thank you for raising this possible area of misunderstanding.

Action: we have added the words 'in deuteropods' to the sentence for clarity. The sentence now reads 'Thus, the evolution of specialised food-capturing deutocerebral appendages in deuteropods could ... ' (line 361)

References cited in response

Averof, M. and Patel, N.H., 1997. Crustacean appendage evolution associated with changes in Hox gene expression. Nature, 388(6643), pp.682-686.

Budd, G.E., 2021. The origin and evolution of the euarthropod labrum. Arthropod structure & development, 62, p.101048.

Lan, T., Zhao, Y., Zhao, F., He, Y., Martinez, P. and Strausfeld, N.J., 2021. Leancoiliidae reveals the ancestral organization of the stem euarthropod brain. Current Biology, 31(19), pp.4397-4404.